# Bipartite binding and partial inhibition links DEPTOR and mTOR in a mutually antagonistic embrace

Maren Heimhalt[1†], Alex Berndt[1†], Jane Wagstaff[1],
Madhanagopal Anandapadamanaban[1], Olga Perisic[1], Sarah Maslen[1],
Stephen McLaughlin[1], Conny Wing-Heng Yu[1], Glenn R Masson[1], Andreas Boland[2],
Xiaodan Ni[1], Keitaro Yamashita[1], Garib N Murshudov[1], Mark Skehel[1],
Stefan M Freund[1], Roger L Williams[1]*

[1]MRC Laboratory of Molecular Biology, Cambridge, United Kingdom; [2]Department of Molecular Biology, University of Geneva, Geneva, Switzerland

**Abstract** The mTORC1 kinase complex regulates cell growth, proliferation, and survival. Because mis-regulation of DEPTOR, an endogenous mTORC1 inhibitor, is associated with some cancers, we reconstituted mTORC1 with DEPTOR to understand its function. We find that DEPTOR is a unique *partial* mTORC1 inhibitor that may have evolved to preserve feedback inhibition of PI3K. Counterintuitively, mTORC1 activated by RHEB or oncogenic mutation is much more potently inhibited by DEPTOR. Although DEPTOR partially inhibits mTORC1, mTORC1 prevents this inhibition by phosphorylating DEPTOR, a mutual antagonism that requires no exogenous factors. Structural analyses of the mTORC1/DEPTOR complex showed DEPTOR's PDZ domain interacting with the mTOR FAT region, and the unstructured linker preceding the PDZ binding to the mTOR FRB domain. The linker and PDZ form the minimal inhibitory unit, but the N-terminal tandem DEP domains also significantly contribute to inhibition.

**\*For correspondence:**
rlw@mrc-lmb.cam.ac.uk

[†]These authors contributed equally to this work

**Competing interests:** The authors declare that no competing interests exist.

## Introduction

The mammalian/mechanistic target of rapamycin complex 1 (mTORC1) is a large (~1 MDa) multiprotein complex consisting of two copies of three subunits: the evolutionary conserved Serine/Threonine protein kinase mTOR (a member of the phosphoinositide-3-kinase-related kinases superfamily of protein kinases, PIKKs), RAPTOR and mLST8 (*Brown et al., 1994*; *Hara et al., 2002*; *Kim et al., 2003*; *Sabatini et al., 1994*; *Sabers et al., 1995*). mTORC1 senses and integrates inputs originating from nutrients, growth factor signaling pathways, oxidative stress, and cellular energy levels (*Jewell et al., 2013*; *Laplante and Sabatini, 2012*). The net response of mTORC1 to such stimuli is to promote cell growth, by activating protein translation, ribosome biogenesis, lipid and nucleic acid biosynthesis and inhibiting autophagy (*Dunlop and Tee, 2014*; *Laplante and Sabatini, 2012*; *Loewith and Hall, 2011*; *Valvezan and Manning, 2019*). The effect of mTORC1 on protein synthesis is primarily due to its phosphorylation of S6 kinase 1 (S6K1) and eIF-4E binding protein 1 (4EBP1) (*Gingras et al., 1999*; *Holz et al., 2005*; *Holz and Blenis, 2005*; *Kang et al., 2013*). Both substrates interact with the mTORC1 complex through their general TOR signaling (TOS) motif that binds the RAPTOR subunit (*Schalm and Blenis, 2002*; *Schalm et al., 2003*; *Schalm et al., 2005*), as well as by forming a second interaction with the FRB helical insertion within the N-lobe of the mTOR kinase domain (*Yang et al., 2017*). A more elaborate mode of mTORC1 substrate interaction has been suggested for the transcription factor TFEB, which lacks a TOS motif, but instead binds an active Rag heterodimer that can form a complex with RAPTOR (*Martina and Puertollano, 2013*;

*Napolitano et al., 2020*). It has been proposed that the substrates' multipartite mode of recognition is crucial for substrate specificity for mTORC1 versus mTORC2 (*Baretić and Williams, 2014*).

In addition to its three core components, mTORC1 associates with various subunits that regulate mTORC1 activity or direct its cellular localization: nutrients such as amino acids promote the association of RAPTOR with heterodimeric Rag GTPases, which, along with the Ragulator complex, recruit mTORC1 to lysosomal surfaces. There, the small GTPase RHEB (Ras homolog enriched in brain) in its GTP-bound state activates mTORC1 via direct interaction with the mTOR catalytic subunit (*Anandapadamanaban et al., 2019*; *Rogala et al., 2019*; *Yang et al., 2017*). Two endogenous negative regulators of mTORC1 activity have been identified to date: DEPTOR (DEP-domain containing mTOR-interacting protein), which also inhibits mTORC2 (*Peterson et al., 2009*), and PRAS40 (proline-rich Akt substrate of 40 kDa) (*Sancak et al., 2007*; *Haar et al., 2007*; *Wang et al., 2007*). PRAS40 interacts with RAPTOR via a TOS-binding motif in a similar fashion as the substrates 4EBP1 and S6K1, resulting in a specificity for mTORC1 (*Yang et al., 2017*). PRAS40 inhibition is lost upon its insulin-stimulated Akt/PKB-mediated phosphorylation, which decreases its affinity for mTORC1 (*Sancak et al., 2007*; *Thedieck et al., 2007*; *Haar et al., 2007*).

Interactions of DEPTOR with the mTOR complexes are less clear. DEPTOR is a 46 kDa protein that consists of three distinct and highly conserved regions (from N- to C-terminus): two tandem DEP domains (a DEPt), an unstructured linker of ~100 residues (residues 228–323, which we have named long-linker) and a C-terminal PDZ domain. The long-linker contains multiple serine phosphorylation sites as well as a consensus βTrCP1-binding site, SSGYFS (referred to as the DEPTOR phosphodegron). Previously, it was suggested that DEPTOR inhibits mTOR function in vivo via an interaction of its PDZ domain with mTOR FAT domain (*Peterson et al., 2009*). However, the nature of this interaction remained elusive.

Under starvation conditions, DEPTOR binds to mTOR and inhibits its kinase activity, whereas under nutrient replete conditions mTOR-dependent phosphorylation at its degron site marks DEPTOR for degradation (*Duan et al., 2011*; *Gao et al., 2011*; *Wang et al., 2012b*; *Zhao et al., 2018*). Two models of DEPTOR-specific mTORC1 regulation were proposed: one suggesting that mTORC1 activation is due to DEPTOR displacement by the phospholipase D1 (PLD1) product phosphatidic acid (PA) (*Yoon et al., 2015*) and another linking mTORC1 inhibition to deubiquitylation of DEPTOR (*Zhao et al., 2018*). Recent reports describe DEPTOR as a negative regulator of mTOR function rather than an inhibitor, as residual mTOR activity is observed in the presence of DEPTOR, in vivo, dampening but not eliminating S6K1 and 4EBP1 phosphorylation (*Caron et al., 2016*; *Dong et al., 2017*; *Hu et al., 2017*; *Laplante et al., 2012*; *Li et al., 2014*). As the mTOR pathway is constitutively activated in cancer, levels of DEPTOR are low in most tumours. Exceptions are a subset of multiple myeloma (MM), thyroid carcinoma, and lung cancers (*Wang et al., 2012a*), where overexpression of DEPTOR eliminates feedback inhibition downstream of mTORC1, resulting in hyperactivation of PI3K/mTORC2/Akt signaling and thereby stimulating cell survival (*Peterson et al., 2009*). RNAi-silencing of overexpressed DEPTOR in a set of MM cells resulted in stalling cell growth and triggering apoptosis, suggesting DEPTOR regulation could be a viable therapeutic strategy (*Peterson et al., 2009*). Indeed, small molecule inhibitors that were recently reported to prevent the formation of an mTOR/DEPTOR complex showed selective cytotoxicity against MM cells (*Lee et al., 2017*; *Shi et al., 2016*; *Vega et al., 2019*). To assist future inhibitor design, we determined the structural and kinetic basis for DEPTOR regulation of mTORC1. The cryo-EM reconstruction of mTORC1 in complex with DEPTOR at 4.3 Å resolution reveals a bipartite binding mechanism of DEPTOR to mTORC1. Specifically, DEPTOR's long-linker and PDZ domain interact with the FRB and FAT domains of mTOR, respectively, and these interactions were validated with NMR and HDX-MS. Kinetic analysis narrowed down DEPTOR's minimal inhibitory unit to the long-linker-PDZ region, revealing that, in contrast to previous cellular studies, the PDZ domain alone is not sufficient for mTORC1 inhibition, and showed that DEPTOR is a unique partial inhibitor. Remarkably, the DEPTOR/mTORC1 complex possesses kinase activity, so that maximal mTORC1 inhibition by DEPTOR leaves the complex with about ~50% residual activity.

## Results

### DEPTOR is a partial inhibitor of mTORC1 in vitro, independent of substrate identity

To characterize the mechanism of mTORC1 inhibition by DEPTOR, we carried out reconstituted inhibition assays with purified recombinant mTORC1, DEPTOR and two major mTORC1 substrates, 4EBP1 (wild-type) and S6K1 (GST-tagged S6K1$^{367-404}$ polypeptide). Using immunoblotting with antibodies specific for the phosphorylated substrates, we found that DEPTOR inhibited mTORC1 with half maximal inhibitory concentration (IC$_{50}$) of 14 µM for wild-type 4EBP1, and 51 µM for S6K1$^{367-404}$ (*Figure 1A*). Interestingly, inhibition of mTORC1 by DEPTOR for both substrates appeared to be partial, with mTORC1 having about 50% residual activity at even the highest concentrations of DEPTOR that could be achieved. This residual activity could also be detected using Phos-tag gels, that efficiently separate phosphorylated from non-phosphorylated proteins (*Figure 1—figure supplement 1A.B*). In contrast, the mTORC1 specific inhibitor PRAS40 showed full inhibition under identical conditions (*Figure 1B*).

Partial enzyme inhibitors bind to the enzyme and decrease its activity, but still allow substrate turnover, even with the inhibitor bound (*Grant, 2018*). Consequently, a partial inhibitor cannot bind in a manner that completely occludes substrate binding. To test whether DEPTOR influences substrate binding, we measured the $K_M$ for 4EBP1 in the presence and absence of DEPTOR (*Figure 1C*). Previously, two distinct $K_{M,4EBP1}$ have been detected, which represent independent binding at the TOS motif binding site on the RAPTOR subunit ($K_{M1}$ = 1.8 µM) and at the FRB-binding site on the mTOR subunit ($K_{M2}$ = 585 µM) (*Figure 1A*; *Yang et al., 2017*). Increasing concentrations of DEPTOR had no apparent effect on the $K_{M,4EBP1}$ of the TOS motif binding site ($K_{M1}$), suggesting that DEPTOR is not able to compete with 4EBP1 at this site (*Figure 1C*). The affinities for the FRB binding were too low to get a reliable kinetic result. Nevertheless, our structural analyses (described below) imply a very similar binding site at the FRB domain for the DEPTOR long-linker and for the 4EBP1/S6K1 substrates, suggesting a partially competitive binding mechanism for this site. The mTORC1 inhibitor PRAS40, which shares the TOS-binding site with 4EBP1, shows full inhibition of 4EBP1 phosphorylation under our assay conditions (*Figure 1B*). DEPTOR's inability to compete at the TOS-motif binding site suggested that inhibition of 4EBP1 phosphorylation may be partial because of unhindered binding of 4EBP1 at the high affinity TOS site located on RAPTOR. To examine this proposition, we employed a TOS-less 4EBP1 mutant as a substrate. DEPTOR seemed to compete more effectively with this mutant, since the IC$_{50}$ was twofold lower with this modified substrate, however, the residual activity remained at about 50% at the maximum DEPTOR concentration (*Figure 1A*). As 4EBP1 binding has been reported to involve additional sites, like the RAIP motif (*Beugnet et al., 2003*; *Eguchi et al., 2006*), we next used a simpler substrate, consisting of a short S6K1 peptide that is expected to exclusively bind to the FRB site (*Yang et al., 2013*). Residual activity in the presence of DEPTOR was also observed with this substrate (*Figure 1A*). Based on these observations, we concluded that partial inhibition by DEPTOR is an intrinsic property of DEPTOR, which takes place independently of substrate identity and substrate binding mode.

To test the possibility that the residual activity is caused by half-site reactivity in mTORC1 via allosteric communication across the dimer interface, we tested DEPTOR inhibition of a monomeric form of mTOR (the mTOR$^{\Delta N}$-mLST8 complex, which lacks the N-terminal 1375 residues of mTOR and the RAPTOR subunit). The IC$_{50}$ for the monomeric mTOR$^{\Delta N}$-mLST8 was 0.4 µM, and the residual activity 33% (*Figure 1D*). Since DEPTOR could still not fully inhibit this monomeric mTOR, we conclude that DEPTOR's residual activity was not caused by effects related to the dimeric nature of mTORC1.

### The minimal inhibitory unit of DEPTOR is the long-linker-PDZ

To determine DEPTOR's mechanism of inhibition of mTORC1 in more detail, DEPTOR deletion variants as well as a 13 S/T→A mutant with most of the phosphorylation sites in DEPTOR removed (*Peterson et al., 2009*) were designed to identify the regions that are crucial for inhibition and the role of DEPTOR phosphorylation in the mechanism (*Figure 2A*). It was reported that DEPTOR's PDZ domain alone (residues 324–409) was sufficient to inhibit mTOR when transiently overexpressed in cells (*Peterson et al., 2009*). However, our in vitro assay showed no measurable inhibition by the PDZ domain alone (*Figure 2B and C*). Furthermore, there was no inhibition by constructs of PDZ

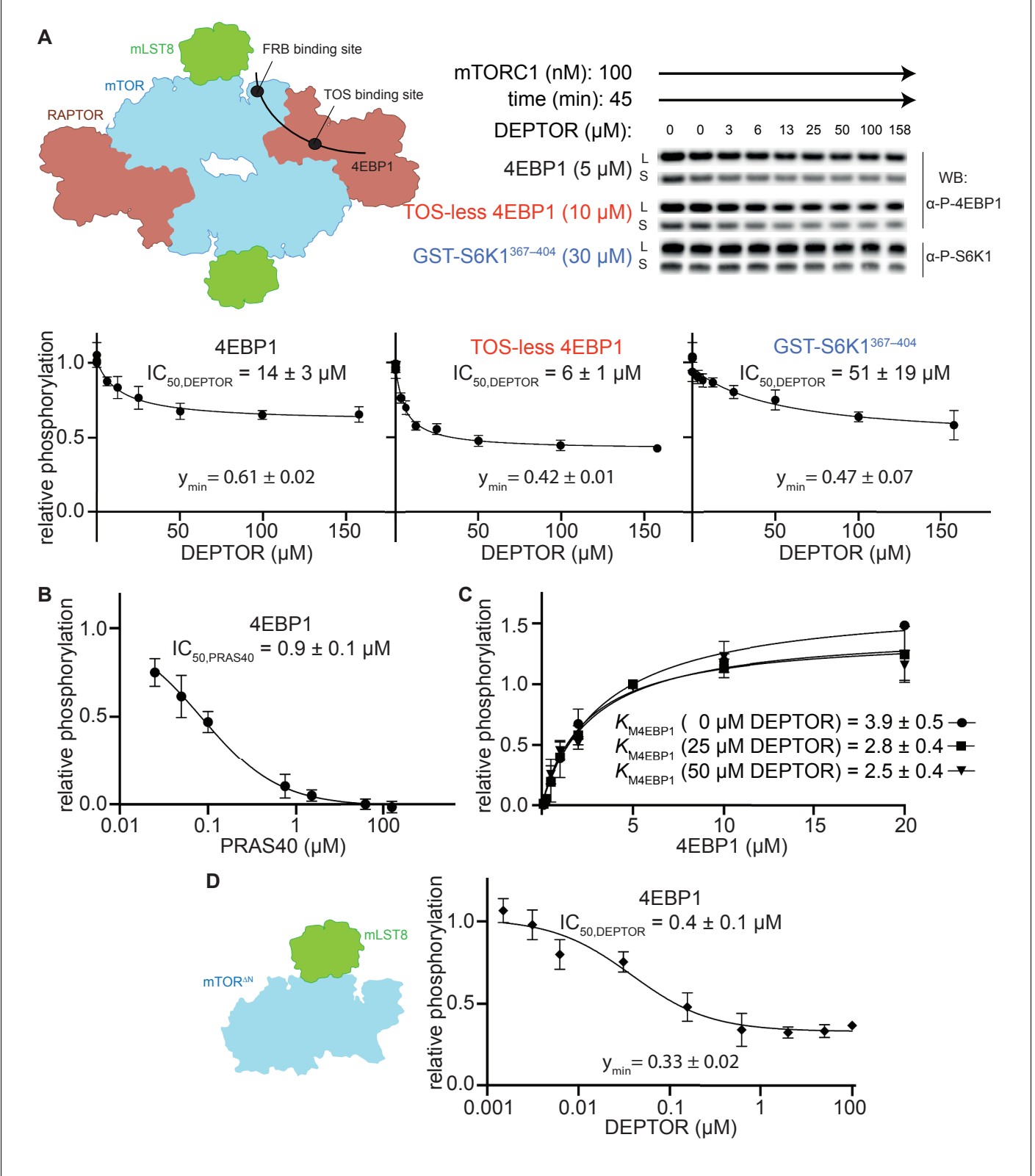

**Figure 1.** DEPTOR is a partial inhibitor of mTORC1, independent of substrate identity. (**A**) Inhibition of the mTORC1 kinase activity by DEPTOR. mTORC1 schematic illustrating the two known substrate-binding sites, the TOS-binding site on RAPTOR and the FRB-binding site on mTOR, for 4EBP1 and S6K1 (left panel). Phosphorylation of substrates was analyzed by western blots using anti p-4EBP1 (Thr37/46) or anti p-S6K1 (Thr389) primary antibodies (right panel). Images of western blots with long (L) and short exposures (S) are shown. The S6K1[367-404] peptide encompasses only the FRB-

*Figure 1 continued on next page*

*Figure 1 continued*

binding site of S6K1. The bottom panels show the quantification of the phosphorylation levels of the substrates based on the western blots from three independent experiments (mean ± SD). Band intensities were normalized to the control (0 µM DEPTOR) and data were plotted and fit by non-linear regression to determine $IC_{50}$ and $y_{min}$ (the residual activity at high [DEPTOR]) as described in Materials and methods. See *Figure 1—figure supplement 1A,B* for complementary experiment using Phos-tag SDS PAGE detection. (**B**) Inhibition of mTORC1 by PRAS40. Inhibition of 4EBP1 phosphorylation is complete under identical reaction conditions as carried out for DEPTOR. (**C**) DEPTOR has no effect on the apparent $K_{M,4EBP1}$. The phosphorylation of 4EBP1 in the absence and presence of DEPTOR (25 µM or 50 µM), normalized to the 5 µM 4EBP1 is plotted (mean ± SD, n ≥ 3) and $K_M$ values were fit as described in Materials and methods. (**D**) Inhibition of monomeric mTOR$^{\Delta N}$-mLST8 (left panel) by DEPTOR (mean ± SD, n ≥ 3). Similar to the wild-type mTORC1 complex, partial inhibition is observed (right panel).

The online version of this article includes the following source data and figure supplement(s) for figure 1:

**Source data 1.** Uncropped blots.
**Source data 2.** Data values for *Figure 1*.
**Figure supplement 1.** DEPTOR is a partial inhibitor of mTORC1, analyzed by Phos-tag SDS PAGE and western blot.
**Figure supplement 1—source data 1.** Uncropped blot and stained gel.
**Figure supplement 1—source data 2.** Data values.

preceded by a short section of the linker (short-linker, described below, *Figure 2—figure supplement 1A*). Unexpectedly, we even observe a slight, but reproducible activation of mTORC1 by these PDZ constructs. In contrast, a DEPTOR construct that includes the entire long-linker region preceding the PDZ *in addition to* the PDZ domain comprises the minimal inhibitory unit. However, the increased $IC_{50}$ for the linker-PDZ construct compared with full-length DEPTOR suggests that the N-terminal tandem DEP domain region (DEPt) contributes to inhibition (*Figure 2C, D*). A mechanism for the inhibitory role of DEPt is suggested by the structural work reported by *Wälchli et al., 2021*. Overexpression of the DEPTOR PDZ domain alone may have an indirect effect on mTOR signaling in cells. Recently, an interaction of DEPTOR PDZ with the C-terminal portion of pREX2, a PTEN inhibiting protein that includes domains structurally related to DEPTOR, has been identified (*Fine et al., 2009*; *Yen et al., 2012*). Considering the intertwined nature of the PI3K/mTOR pathways with numerous feedback loops and crosstalk modes, it is difficult to unambiguously identify the origin of mTOR inhibition by the PDZ domain of DEPTOR in cell-based experiments. The tandem DEP domains alone (residues 1–221) or in combination with the long-linker (residues 1–323) showed no inhibition (*Figure 2B and C*).

DEPTOR is phosphorylated in an mTOR-dependent manner in cells, and phosphorylation sites have been identified in the long-linker region (*Duan et al., 2011*; *Gao et al., 2011*; *Peterson et al., 2009*; *Wang et al., 2012b*; *Zhao et al., 2018*). To check the possibility that the inhibition of 4EBP1 and S6K1 phosphorylation by DEPTOR is due solely to DEPTOR acting as an alternative substrate for mTORC1, we tested inhibition of mTORC1 by a previously described DEPTOR mutant in which 13 S/T residues in the long-linker region were mutated to alanine (13A mutant) (*Peterson et al., 2009*). The $IC_{50}$ of this DEPTOR mutant was comparable to the wild-type DEPTOR for both the wild-type mTORC1 (*Figure 2D*) and mTORC1 with an activating cancer-associated mTOR mutation (A1459P, *Figure 2—figure supplement 1C*), so interaction between these S/T residues and mTORC1 is not important for inhibition by DEPTOR. The 13A mutant is a partial inhibitor, like wild-type DEPTOR, meaning that they bind to mTORC1 without preventing substrate binding and turnover.

We next assayed phosphorylation of full-length DEPTOR and its deletion variants via Phos-tag SDS PAGE. To increase the signal of phosphorylated DEPTOR for reliable detection, we used the hyperactive mTORC1 A1459P. The isolated PDZ as well as tandem DEP domains were not phosphorylated at any detectable rate in an mTOR dependent manner (*Figure 2—figure supplement 1D*), while the DEPDEP-long-linker was phosphorylated at a reduced rate compared to wild-type DEPTOR at equimolar concentrations (*Figure 2—figure supplement 1E*). This is an additional indication that the PDZ domain assists the DEPTOR long-linker interaction with mTORC1.

## Cryo-EM structure of mTORC1/DEPTOR reveals a bipartite binding mode of DEPTOR to mTOR

We determined the structure of mTORC1 bound to full length DEPTOR by electron cryo-microscopy (cryo-EM). Using cross-linked mTORC1/DEPTOR, we generated a 4.3 Å resolution reconstruction of

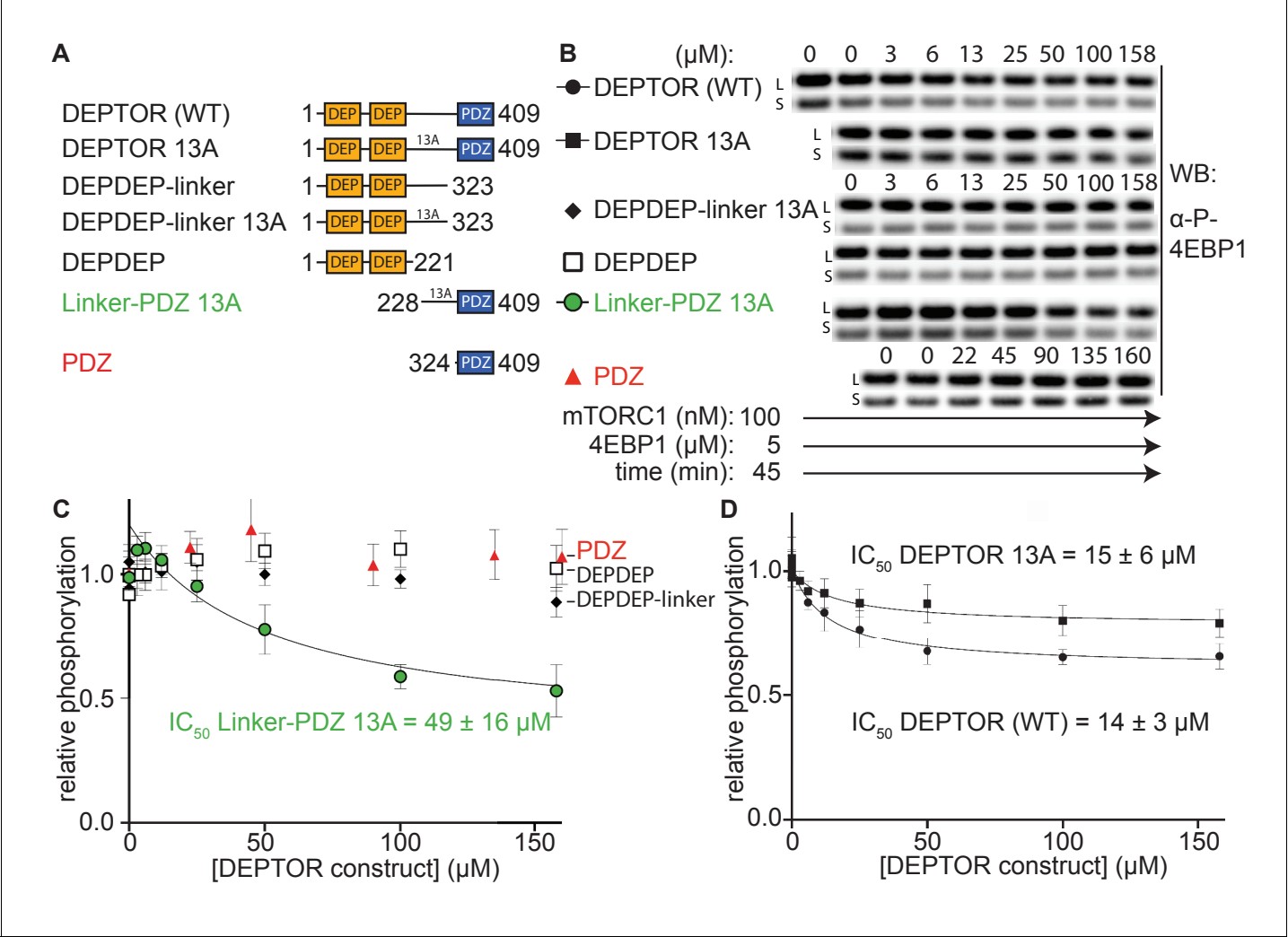

**Figure 2.** The minimal inhibitory unit of DEPTOR is long-linker-PDZ. (**A**) DEPTOR deletion variants tested as inhibitors and substrates for mTORC1. (**B**) Immunoblots showing the residual phosphorylation of 4EBP1 in the presence of various DEPTOR deletion variants. Images of western blots with long (L) and short exposures (S) are shown. (**C**) and (**D**) quantification of western blots of phosphorylated 4EBP1 plotted as a fraction of the control (0 μM inhibitor) vs. inhibited (mean ± SD, n ≥ 3) and fit to a non-linear regression for all deletion variants to determine IC$_{50}$. While all experiments were performed at 30°C, inhibition of mTORC1 by the PDZ domain was tested at 20°C for 20 min as the domain stability was low. N-terminally extended PDZ constructs that showed increased temperature stability showed no inhibition of mTORC1 (**Figure 2—figure supplement 1A**). To demonstrate that the temperature difference had no effect on the inhibition, DEPTOR (WT) was tested at 20°C for mTORC1 inhibition (**Figure 2—figure supplement 1B**). The data shown for DEPTOR (WT) is also part of **Figure 1A**. DEPTOR 13A inhibition of the mTORC1 A1459P mutant is shown in **Figure 2—figure supplement 1C**.

The online version of this article includes the following source data and figure supplement(s) for figure 2:

**Source data 1.** Uncropped blots.

**Source data 2.** Data values for **Figure 2**.

**Figure supplement 1.** DEPTOR is a partial inhibitor of mTORC1, analyzed by Phos-tag SDS PAGE and western blot.

**Figure supplement 1—source data 1.** Uncropped stained gels.

**Figure supplement 1—source data 2.** Data values.

mTORC1 in a complex with DEPTOR (**Figure 3A**). The overall architecture of the mTORC1 complex bound to DEPTOR resembles the conformation of mTORC1 in the absence of RHEB (PDB 6BCX). However, an additional density can be observed in a crevice between the FAT domain and the N-heat of mTOR, centred at residues 1527–1571 in the FAT domain. The shape and size of this extra density is consistent with the structure of a PDZ domain (**Figure 3C**).

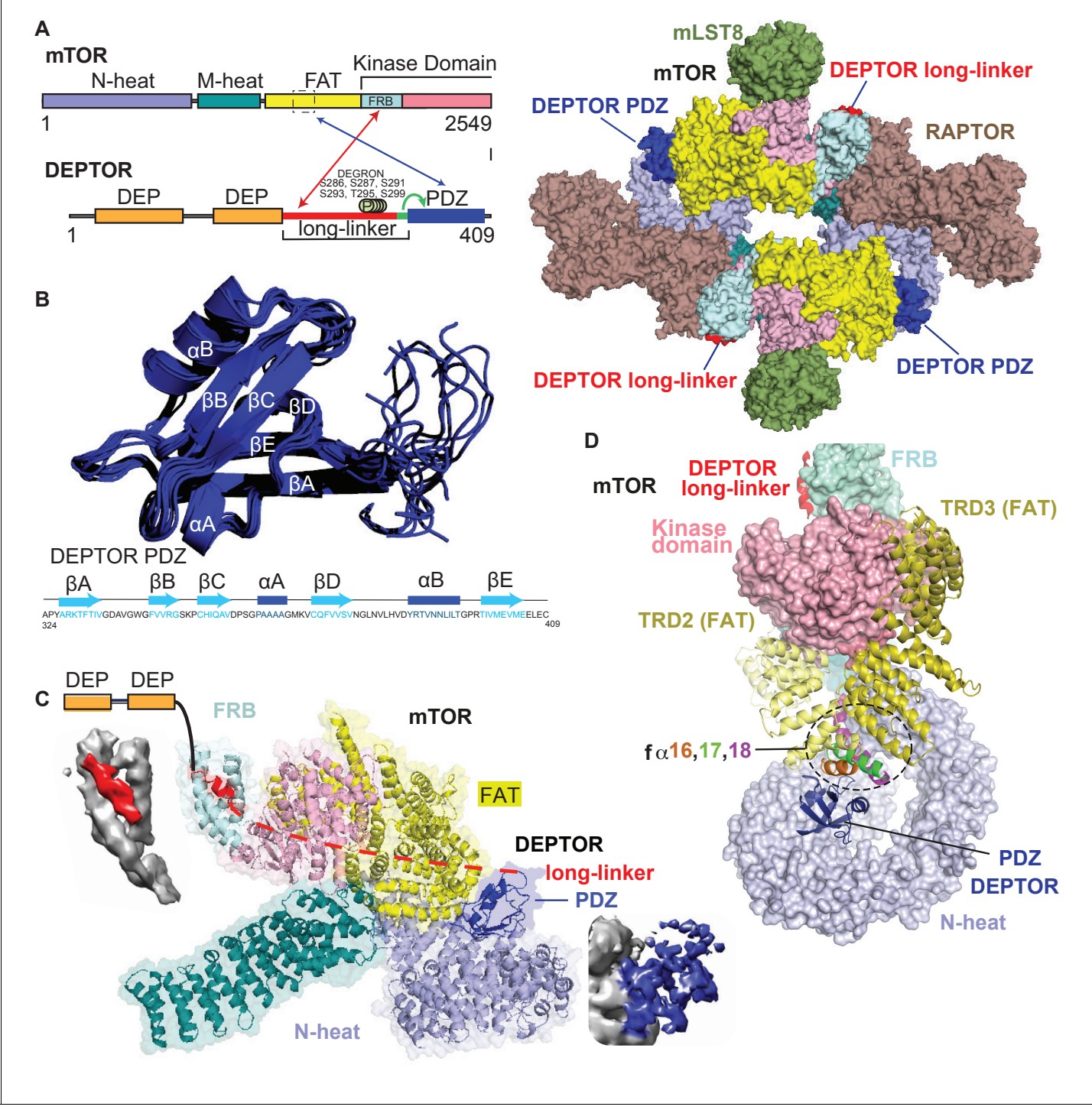

**Figure 3.** Cryo-EM structure of mTORC1/DEPTOR reveals bipartite binding mode of DEPTOR linker-PDZ to mTOR. (**A**) Domain organization of mTOR and DEPTOR is shown on the left. The region of interaction between the mTOR FAT domain residues 1527–1571 and the DEPTOR PDZ domain, as well as the mTOR FRB domain and the DEPTOR linker are highlighted with arrows. Surface representation of the model for the mTORC1/DEPTOR complex is shown on the right, color-coded by domains. Only the linker and PDZ domain, not the tandem DEP domain of DEPTOR are visible in the cryo-EM density. See *Figure 3—figure supplement 2* and *Table 1* for cryo-EM data details. (**B**) The 10 lowest energy homology models of DEPTOR PDZ produced using CS-Rosetta guided by NMR data, including 14 NOE distance restraints. DEPTOR PDZ construct (324-409) was used. See *Figure 5— figure supplement 2A* for the assigned $^{1}$H-$^{15}$N BEST-TROSY spectrum of the PDZ domain. (**C**) A close-up of DEPTOR binding to the mTOR subunit shows DEPTOR PDZ domain in a crevice between the FAT domain and the N-heat of mTOR. DEPTOR long-linker (red) forms interactions at the FRB domain. The dashed line spans the distance of DEPTOR long-linker between the two binding sites. Cryo-EM density for DEPTOR and its binding sites is

*Figure 3 continued on next page*

*Figure 3 continued*

shown next to the model. The density for mTOR, PDZ, and long-linker are colored in gray, blue, and red, respectively. (**D**) Three mTOR helices (fα16, fα17, fα18) at the junction of two solenoids (TRD2 and TRD3) in the FAT domain are splayed and form a non-canonical interface with the DEPTOR PDZ domain.

The online version of this article includes the following figure supplement(s) for figure 3:

**Figure supplement 1.** DEPTOR interaction with mTORC1.

**Figure supplement 2.** Cryo-EM data of the mTORC1/DEPTOR complex.

The density for the DEPTOR PDZ domain was poorly resolved. As crystallization of the 324–409 construct remained unsuccessful, the structure of the DEPTOR PDZ domain was determined using NMR spectroscopy (*Figure 3B*). Here, instead of calculating a formal solution structure of the PDZ domain, NMR restraint-based models were created using the ROSETTA software suite. The assigned backbone chemical shifts, which are uniquely sensitive to the secondary structure in which each residue resides, were employed to more accurately select fragments of homologous proteins using the POMONA webserver. These fragments then were used to calculate the PDZ model using CS-RosettaCM that combines both chemical shifts (CS) and comparative modeling (CM) (*Shen and Bax, 2015*). An ensemble of structures was then further refined using a limited set of NOE distance restraints, resulting in a homology model that also satisfies solution state parameters that we observed. This process is advantageous as it creates data driven models significantly faster than traditional NMR structure calculation techniques. *Figure 3B* represents an ensemble of the 10 lowest energy structures calculated using this methodology, with the lowest energy model used in the final cryo-EM structure refinement. The PDZ domain binds to three consecutive helices (fα16, fα17, fα18, residues 1525–1580), in the middle of the C-shaped FAT domain at a corkscrew junction between two helical solenoids (previously referred to as TRD2 and TRD3 *Yang et al., 2013*; *Figure 3C and D*). The key interactions in this surface are formed by E1530, E1531, C1534, M1535, R1538, and Q1562 on the mTORC1 FAT domain, and residues 336–342 and 354–362 on the DEPTOR PDZ domain.

We identified a second density adjacent to the mTOR FRB domain, covering the lipophilic patch formed by the FRB domain residues Y2038, F2038, Y2105, and F2108 (*Figure 3C and D*). We modeled the density as three turns of an α-helix, but the local resolution is insufficient to assign a sequence. It is plausible that this density represents part of the DEPTOR long-linker preceding the PDZ domain, since our kinetic results identified the long-linker-PDZ as the minimal region of DEPTOR required for mTORC1 inhibition. However, we also see density at the TOS and RAIP-binding sites on RAPTOR, raising the possibility that an unidentified substrate might be associated with the mTORC1, although no substrate was included in the sample preparation. To verify the interaction of DEPTOR with the FRB, we carried out a series of experiments described below. Neither the long-linker nor the PDZ interactions alone are sufficient for the inhibition by DEPTOR (*Figure 2C*). Our structure suggests a bipartite DEPTOR binding mode, with one interaction involving the FRB, similar to the mTORC1 substrates S6K1 and 4EBP1, while the other interaction involves the DEPTOR PDZ domain binding to mTORC1's FAT domain and is unique to DEPTOR. The PDZ binding site is remote from the active site and serves as an anchor moiety (*Figure 3C*), similarly to the TOS motif of the substrates such as 4EBP1 that anchor to the TOS-binding site on the RAPTOR subunit. We cannot discern any density spanning the two DEPTOR binding sites, and HDX-MS suggests that the long-linker is unstructured (*Figure 3—figure supplement 1A*, *Supplementary file 1*). Furthermore, an NMR-based secondary structure analysis of the isolated DEPTOR long-linker chemical shifts revealed no regions of secondary structure (*Figure 3—figure supplement 1B*). Consequently, it is impossible to say whether DEPTOR spans the two mTOR binding sites on one molecule, on different mTOR subunits in the dimer or a combination of both of these modes (*Figure 3—figure supplement 1C*).

When the structures of free mTORC1 (PDB 6BCX), RHEB-bound mTORC1 (PDB 6BCU) and mTORC1 bound to DEPTOR are aligned locally on the C-lobe of the kinase domain, it is clear that the ATP-binding sites of the free mTORC1 and DEPTOR-bound mTORC1 are very similar and both are distinct from the RHEB-bound conformation (*Figure 3—figure supplement 1D*). Free mTORC1 is present in a continuum of conformations, ranging from an open conformation to a closed

conformation that are both very different than the RHEB-bound, activated conformation (*Yang et al., 2017*). It may be that the predominant conformation captured by cryo-EM for free mTORC1 corresponds to an inhibited conformation that is only capable of slowly turning over substrate and this same conformation is the one predominately bound to DEPTOR.

## DEPTOR PDZ domain binds to mTORC1 in a non-canonical manner

Typically, PDZ domains bind their targets via the C-terminal tail of the target protein binding in the PDZ αB/βB binding groove (*Ernst et al., 2014*). The cryo-EM density for the DEPTOR PDZ domain bound to mTOR suggests that binding occurs in a non-canonical fashion. The PDZ domain binds to helices in the mTOR FAT domain (*Figure 3D*). This unique type of interaction was further examined by an NMR-binding experiment using the non-labeled 1 MDa mTORC1 as a binding partner for the isolated $^2$H,$^{13}$C,$^{15}$N DEPTOR PDZ domain. Large protein complexes like mTORC1 would generally severely impair NMR studies due to increased linewidth caused by slower tumbling. The same would apply for any labeled interactor that binds tightly to the complex and shows a slow dissociation rate. Nonetheless, in case of a $K_d$ in the range of 0.1 μM to 1 mM and an excess of the interactor in the NMR experiment, the effects of binding can be imprinted onto the dissociated small interactor (*Maurer et al., 2004*). In this case, binding effects on the small interactor can be observed in the bulk unbound and labeled protein. The DEPTOR PDZ/mTORC1 interaction showed ideal properties for this experiment and DEPTOR PDZ binding to mTORC1 could be successfully detected from the pool of free PDZ domain. Based on the line broadening, DEPTOR PDZ residues 338–342 and 358–362 form the interface with mTORC1 (*Figure 4A*, *Figure 4—figure supplement 1*). While the region 338–342 involves the canonical binding residues, the region 358–362 is part of the PDZ βC/αA structural element and therefore outside the canonical αB/βB binding groove. The binding interface on DEPTOR PDZ is surprisingly similar to the binding mode of the third PDZ domain of the scaffold protein inactivation-no-afterpotential D (INAD) to the TRP channel in *Drosophila* photoreceptors (*Figure 4B*), indicating that although this is a rare mode of interaction, it is not unprecedented (*Ye et al., 2016*).

## A set of linker regions bind to mTOR's FRB domain

Because our cryo-EM structure showed a bipartite binding that included the FRB domain, and because the PDZ domain alone is insufficient for mTORC1 inhibition, we attempted to define the region of DEPTOR interacting with mTOR's FRB. A small stretch of extra cryo-EM density in mTOR's FRB domain suggested an interaction of DEPTOR with this mTOR domain (*Figure 3C*). As it has been previously reported that DEPTOR's long-linker is phosphorylated in an mTOR-dependent manner (*Peterson et al., 2009*), we proposed that a region in the DEPTOR long-linker binds to the FRB. To verify this interaction, binding of the long-linker-PDZ (residues 228–409) to the isolated FRB domain was characterized by NMR. The chemical shift perturbation in the $^1$H-$^{15}$N BEST-TROSY NMR experiments comparing the bound FRB vs. free FRB confirmed that the long-linker interacts with the FRB domain (*Figure 5A*, *Figure 5—figure supplement 1A–C*). The two most prominent patches affected by this interaction are located at the FRB residues 2035–2042 in kfα1 and 2103–2109 in kfα4, which is consistent with the cryo-EM density (*Figure 5A*). This suggests a similar binding location of the DEPTOR linker at the FRB as the substrates S6K1 or 4EBP1 (*Yang et al., 2017*). Similar to the substrates, binding affinity of the DEPTOR long-linker to the FRB is weak, with a $K_d$ estimated by NMR to be ~500 μM.

To map the regions in DEPTOR interacting with the FRB domain, we carried out NMR experiments with the isolated DEPTOR long-linker and HDX-MS experiments with the full-length DEPTOR. The chemical shift perturbation of the DEPTOR long-linker (residues 228–323) bound to FRB vs. DEPTOR long-linker alone in $^1$H-$^{15}$N BEST-TROSY NMR experiments revealed four patches, each of five residues in the long-linker that were altered by FRB binding, residues 228–232, 244–249, 261–264, and 280–285 (*Figure 5B*). Indeed, the regions of increased protection observed in full-length DEPTOR in the presence of the FRB in an HDX-MS experiment were in good agreement with the NMR result, as DEPTOR peptides 230–248, 270–288 and 276–296 were protected from solvent exchange in the presence of the FRB (*Figure 5C*, *Supplementary file 2*). Notably, both experiments suggested that there was not one unique FRB-binding motif in the long-linker, but multiple regions capable of interacting. This is not surprising, as it has previously been shown for the mTORC1

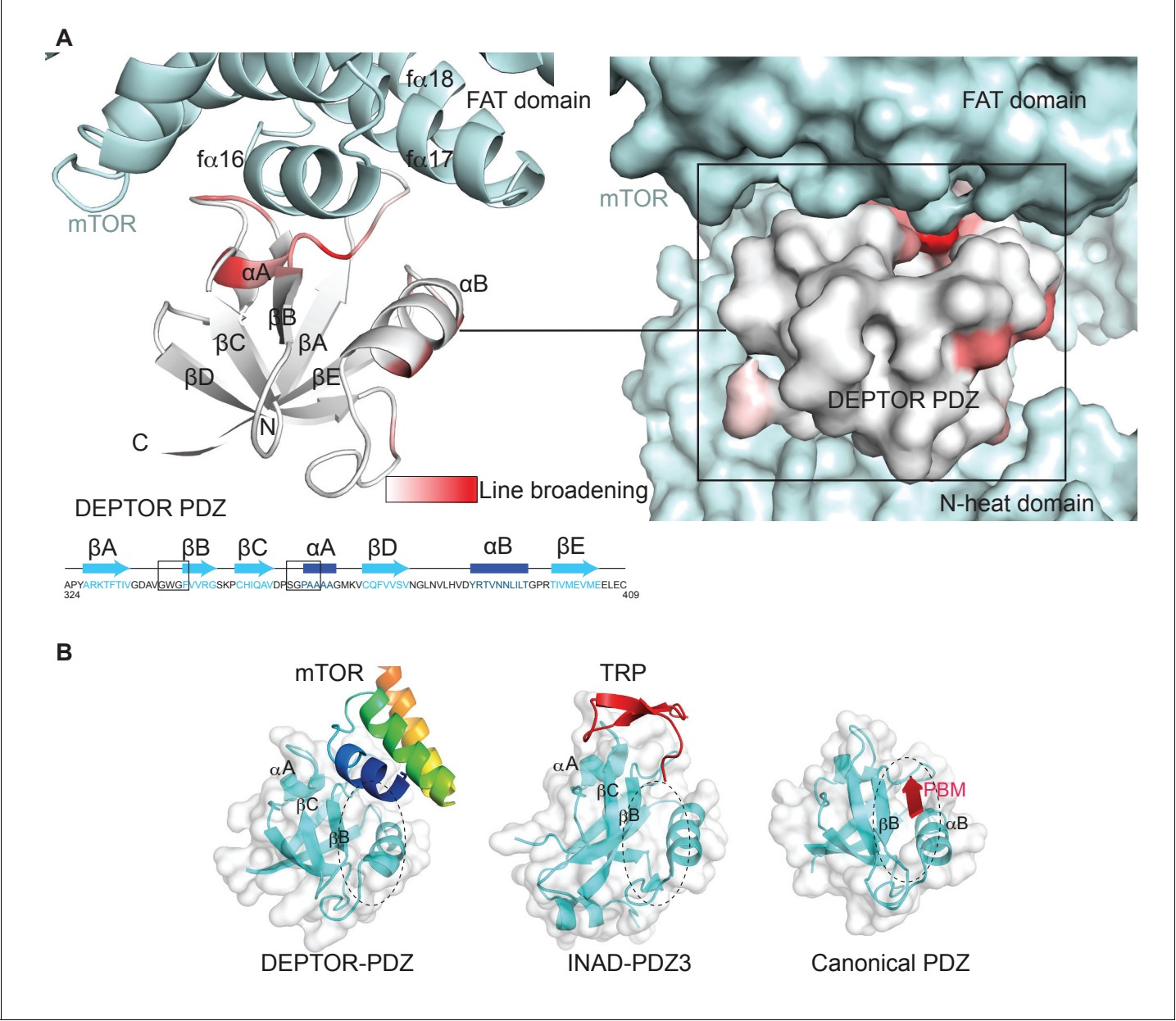

**Figure 4.** DEPTOR PDZ domain binds to mTORC1 in a non-canonical manner. (**A**) Line broadening of isotope-labeled DEPTOR PDZ (residues 324–409) caused by its binding to the full-length mTORC1 is displayed in shades of red on the ribbon diagram of the PDZ domain. The NMR data is consistent with the non-canonical binding mode of DEPTOR PDZ seen with the cryo-EM. See *Figure 4—figure supplement 1* for NMR differential line-broadening analysis. (**B**) A comparison of canonical and non-canonical binding to PDZ domains. The canonical binding mode (PDB 1BE9) is illustrated by the third PDZ domain of postsynaptic density-95 (PSD-95) interacting with the cysteine-rich interactor of PDZ3 (CRIPT). The DEPTOR PDZ binds to the mTOR FAT region in a non-canonical manner. Distinct non-canonical binding is also seen for other PDZ domains, such as the interaction of PDZ3 of inactivation no afterpotential D (INAD) with the transient receptor potential (TRP) channel (PDB 5F67).

The online version of this article includes the following figure supplement(s) for figure 4:

**Figure supplement 1.** DEPTOR interaction with mTORC1.

substrate 4EBP1 that the FRB-interacting region varies based on the substrate's individual phosphorylation sites (*Yang et al., 2017*). This most likely also applies to DEPTOR, and many mTOR-dependent phosphorylation sites have previously been reported in DEPTOR's long-linker (*Duan et al., 2011*; *Gao et al., 2011*; *Peterson et al., 2009*). In the HDX-MS experiment, there is also protection in the DEPTOR N-terminal region (peptide 2–32 protected) that includes part of the DEPt. Because

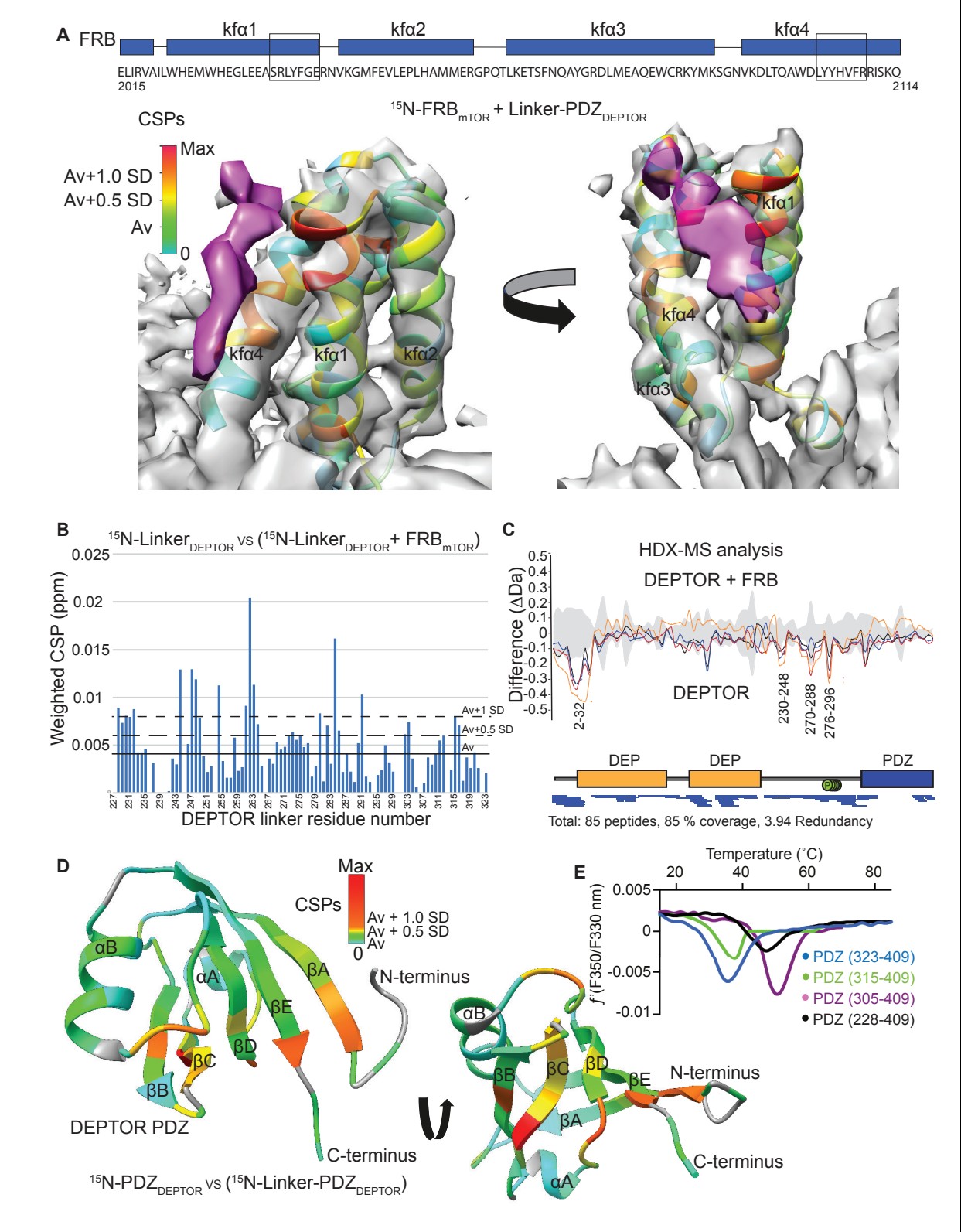

**Figure 5.** DEPTOR linker has two parts, each with a distinct role. (**A**) Mapping the mTOR FRB surface that binds to the DEPTOR linker by the chemical shift perturbations (CSPs) of the FRB (residues 2015–2114) bound to DEPTOR linker-PDZ (residues 228–409) vs. free FRB in a $^1$H-$^{15}$N BEST-TROSY experiment. The CSPs are displayed on the ribbon model of mTORC1 that is shown together with the EM density (semitransparent gray) for the FRB region of the mTORC1/DEPTOR complex. The EM density corresponding to DEPTOR is colored purple. See *Figure 5—figure supplement 1A–C* for

*Figure 5 continued on next page*

*Figure 5 continued*

NMR spectra and analysis. **(B)** Mapping the DEPTOR linker regions that interact with the FRB domain by the chemical shift perturbation of the FRB-bound linker vs. free linker in a $^1$H-$^{15}$N BEST-TROSY experiment. See *Figure 5—figure supplement 1D* for NMR spectra. **(C)** Mapping the DEPTOR regions interacting with the mTOR FRB investigated by HDX-MS. Reduction in HDX in full-length DEPTOR in the presence of the FRB suggests that regions of the linker interact with the FRB. Four different colored lines represent four time points (3 s = orange, 30 s = red, 300 s = blue, 3000 s = black), the gray colored area represents the standard error. See also *Supplementary file 2*. **(D)** Interaction of the PDZ domain with a portion of a DEPTOR linker revealed from a comparison of DEPTOR PDZ (residues 324–409) and linker-PDZ (residues 228–409) $^1$H-$^{15}$N BEST-TROSY spectra. The weighted chemical shift perturbation is calculated between the nearest linker-PDZ peak to the assigned PDZ peak in the overlaid spectra (see *Figure 5—figure supplement 2A*). The minimal map colored from green to red shows the interaction of the linker with the surface of the PDZ. **(E)** Stability of long-linker-PDZ (residues 228–409) and short-linker-PDZ (305-409) is greatly improved over the PDZ domain alone (residues 324–409), as shown by ~10℃ increase in the protein melting temperature measured by differential scanning fluorimetry.

The online version of this article includes the following source data and figure supplement(s) for figure 5:

**Source data 1.** Data values for *Figure 5*.
**Figure supplement 1.** Analysis of the mTOR FRB domain and DEPTOR linker interaction by NMR.
**Figure supplement 2.** DEPTOR PDZ domain is greatly stabilized by its interaction with a short stretch of DEPTOR linker.

the linker-PDZ is sufficient for the inhibition of mTORC1, only this region was included in our NMR-based study of the interaction of DEPTOR with the FRB. Interestingly, a structure of the tandem DEP domains that was recently reported, which included a small segment (residues 228–235) of the long linker, showed that this segment forms an α-helix that interacts with the N-terminus of the DEP domain (residues 21–32), in a pocket between the two DEP domains (*Weng et al., 2021*). It is possible that the protection that we see in the N-terminal region of DEPTOR upon binding the FRB is due to an influence of the FRB on the rigidity of the long-linker, thereby stabilizing the interaction between the linker and the tandem DEP domain region.

All FRB interacting areas in the DEPTOR linker are within 45 residues of the PDZ domain, which could cover a distance of about 160 Å, if in an extended conformation. As most of the linker showed no indication of secondary structure elements, this length should be sufficient for a bipartite mTORC1 binding interaction involving the DEPTOR PDZ and the long-linker binding either a single mTOR subunit or across the dimer interface (*Figure 3—figure supplement 1C*).

## DEPTOR PDZ forms an interaction with the long-linker that controls domain stability and creates a unique surface on PDZ

Secondary chemical shift analysis of the isolated DEPTOR long-linker suggested that the long-linker has no residual secondary structure (*Figure 3—figure supplement 1B*). However, the linker unexpectedly contributes significantly to the stability of the PDZ domain by interacting along the PDZ surface, following a patch of mostly uncharged and hydrophobic residues. This interaction could be detected due to a chemical shift perturbation for some PDZ domain residues in the $^1$H-$^{15}$N BEST-TROSY spectrum for a construct consisting of the long-linker PDZ when compared to the $^1$H-$^{15}$N BEST-TROSY spectrum of the PDZ domain alone (*Figure 5D*, *Figure 5—figure supplement 2A*). By N-terminal deletion analysis of the long-linker-PDZ, we identified a small portion of the linker that is sufficient to stabilize the PDZ domain: a construct consisting of residues 305–409 (here referred to as short-linker PDZ) showed an increased melting temperature by about 10℃ (*Figure 5E*). Furthermore, NMR $^{15}$N backbone relaxation experiments including $T_1$, $T_2$ and $^{15}$N{$^1$H} heteronuclear NOE analyses showed that the PDZ alone (PDZ-only, residues 324–409) has a high flexibility in the N-terminal region, encompassing the first β strand and the subsequent loop (*Figure 5—figure supplement 2B*). This suggests that the short-linker region significantly adds to the overall stability of the PDZ domain, but the short-linker-PDZ is not an mTORC1 inhibitor (*Figure 2—figure supplement 1A*).

## DEPTOR inhibits activated mTORC1 more strongly than basal mTORC1

Our structural analysis showed that DEPTOR PDZ binds close to a region on the mTOR FAT domain that undergoes a major conformational change induced by the RHEB-GTP binding (*Yang et al., 2017*). This suggested that DEPTOR inhibition could be altered as a result of activation. To determine the effect of DEPTOR on activated mTORC1 in our reconstituted system, we tested DEPTOR inhibition of RHEB-GTP- or mutation-activated mTORC1. Surprisingly, the $IC_{50}$ for DEPTOR inhibition

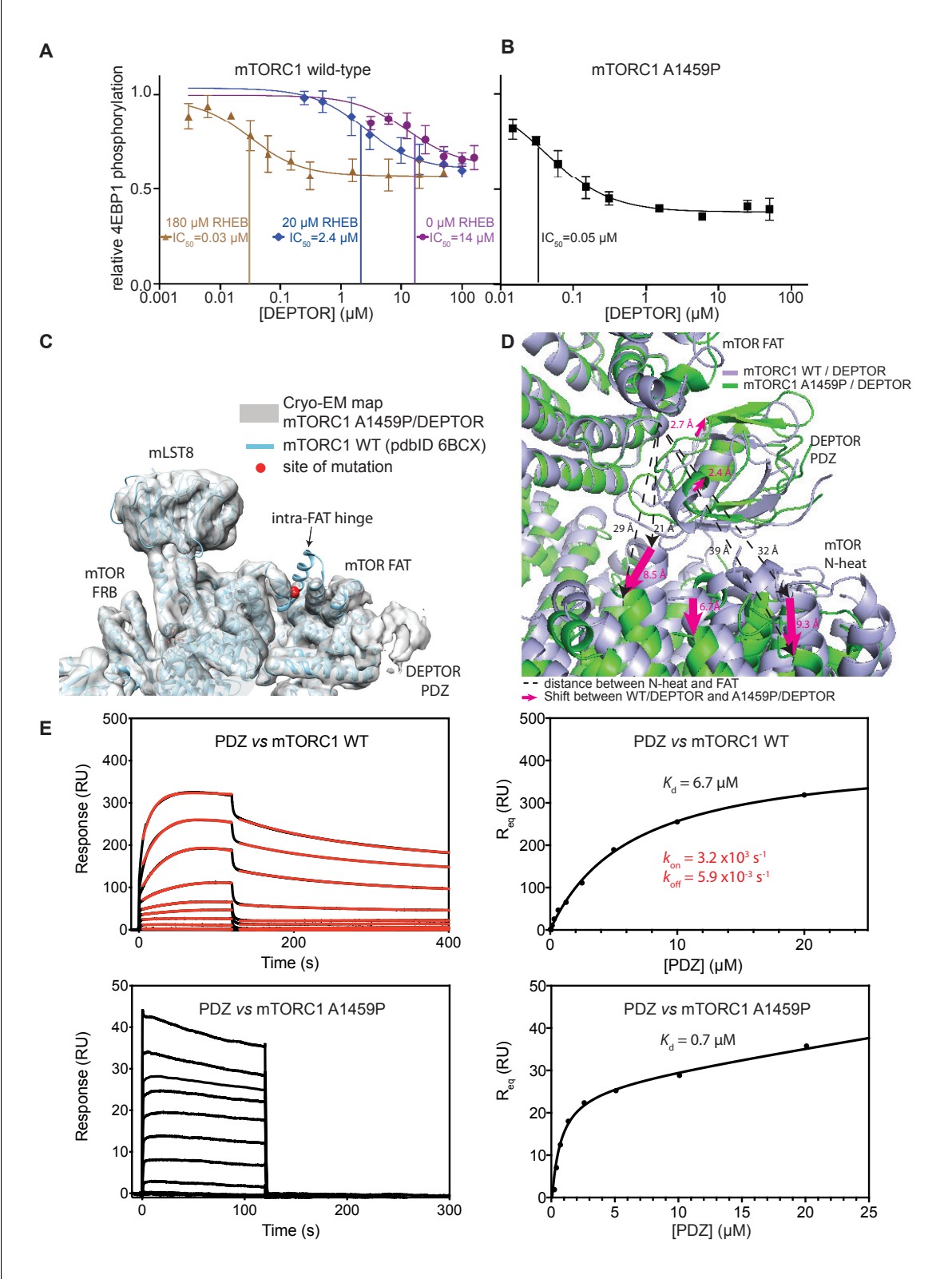

**Figure 6.** DEPTOR inhibits activated mTORC1 more strongly than basal mTORC1, but phosphorylated DEPTOR does not inhibit mTORC1. (**A**) Increasing concentrations of RHEB-GTP lead to activated mTORC1 and at the same time result in a decreased DEPTOR $IC_{50}$ (substrate was 4EBP1). Band intensities reflecting P-4EBP1 were normalized to the control (0 µM DEPTOR) and the data (mean ± SD, n ≥ 3) was fit by a nonlinear regression to determine $IC_{50}$. (**B**) Cancer-associated, hyperactive mutant mTORC1-A1459P also shows a decreased DEPTOR $IC_{50}$. Band intensities reflecting P-4EBP1

*Figure 6 continued on next page*

*Figure 6 continued*

were normalized to the control (0 μM DEPTOR) and the data (mean ± SD, n ≥ 3) was fit by a nonlinear regression to determine $IC_{50}$. (C) Cryo-EM reconstruction of A1459P mTORC1 mutant and DEPTOR reveals the loss of an mTOR helix at the mutation site. The site of mutation lies within the hinges of mTORC1 which are involved in introducing major conformational changes in mTOR upon RHEB-induced activation. (D) Alignment of WT and mutant A1459 mTORC1 on the PDZ-binding site in the FAT domain (shown with green density bound to the PDZ domain illustrated with magenta spheres) reveals a shift of mTOR N-heat domain for mTORC1 A1459P/DEPTOR (yellow density) with respect to mTORC1 WT/DEPTOR (gray density). This shift increases the crevice between the FAT and the N-heat domains in mTORC1 A1459P and creates an easier access for DEPTOR PDZ to its binding site. (E) Comparison of PDZ binding to wild-type and mutant mTORC1 analyzed by SPR. PDZ (construct 305–409) binds to wild-type mTORC1 with slow on/off kinetics and lower affinity, whereas it binds the mutant mTORC1, with fast on/off kinetics and 10-fold greater affinity. The total binding data were fit to a model with one-site specific binding combined with a linear non-specific component.

The online version of this article includes the following source data and figure supplement(s) for figure 6:

**Source data 1.** Data values for *Figure 6*.
**Figure supplement 1.** DEPTOR's inhibition of activated mTORC1.
**Figure supplement 1—source data 1.** Uncropped stained gel.
**Figure supplement 1—source data 2.** Data values.
**Figure supplement 2.** Cryo-EM data of the mTORC1 A1459P/DEPTOR complex.

---

decreased about 400-fold in the presence of RHEB-GTP compared to mTORC1 alone (*Figure 6A*), suggesting tighter binding of DEPTOR to the activated mTORC1. A similar reduction in DEPTOR $IC_{50}$ was observed for the cancer-associated mTOR mutant A1459P (*Figure 6B*). In contrast, the $EC_{50}$ for RHEB-GTP activating wild-type mTORC1 was not significantly affected by the presence of DEPTOR (*Figure 6—figure supplement 1A*). Cancer-associated mutations clustering around C1483 were shown to involve structure-stabilizing residues (*Yang et al., 2017*). These mutations significantly lower the activation energy for the transition from ground state to the RHEB-GTP-bound activated state (*Yang et al., 2017*). The cancer-associated mutant A1459P is in the middle of a helix fα12 at the major intra-FAT hinge, and our cryo-EM structure of the A1459P mutant bound to DEPTOR (global resolution 4.7 Å) shows this helix becomes disordered (residues 1457–1470) (*Figure 6C*). This disorder likely increases plasticity of the intra-FAT hinge, thereby improving DEPTOR binding. The structure of mutant mTORC1 bound to DEPTOR reveals a widening of the cleft between the mTOR N-heat and the PDZ-bound FAT domain, relative to this cleft in the structure of DEPTOR bound to wild-type mTORC1 (*Figure 6D*). This cleft widening might enable easier access for DEPTOR PDZ to bind to the FAT domain. Consistent with this proposal, we observe fast on/off kinetics for PDZ binding to the mTORC1 mutant A1459P with a $K_d$ of 0.6 μM determined by SPR (*Figure 6E*). For the wild-type mTORC1, the $K_d$ is increased ten-fold ($K_d$ = 7 μM), and we observe slow on/off kinetics, which suggests that there is a rate-limiting conformational change within wild-type mTOR to facilitate binding of PDZ. Furthermore, a multibody refinement of the mTORC1/DEPTOR cryo-EM data set indicates that one of the major motions in the wild-type mTORC1/ DEPTOR complex lies in the N-heat domain (*Figure 6—figure supplement 1B*). The proposition that the N-heat in wild-type mTOR restricts DEPTOR PDZ binding is further supported by the 40-fold lowered $IC_{50}$ for DEPTOR inhibition of the mTOR^{ΔN}-mLST8 (*Figure 1D*), which does not possess the N-heat domain. Structurally, the interaction of DEPTOR PDZ with the FAT domain does not seem to be significantly altered between mutant and wild-type mTORC1. Despite tight binding of DEPTOR PDZ to the A1459P mutant, there is no inhibition of the mutant by the isolated PDZ domain (without the long linker), even at saturating concentrations (*Figure 6—figure supplement 1C*). This underlines the need for a bipartite interaction for effective mTORC1 inhibition by DEPTOR.

## Phosphorylated DEPTOR does not inhibit mTORC1

In order to reconcile our kinetic and binding results suggesting that a cancer-associated, activated mTOR mutant shows increased DEPTOR association, with previous results showing a *decrease* in the amount of DEPTOR co-immunoprecipitated with exogenously expressed mutant mTOR, we examined the effect of sustained mTORC1 activity on the association of DEPTOR with mTORC1. In our kinetic analysis of the effect of DEPTOR on phosphorylation of S6K1 and 4EBP1 above, we used initial rates of mTORC1 before significant substrate depletion. However, under these conditions, there is also very little phosphorylated DEPTOR produced. To investigate whether mTOR-dependent phosphorylation of DEPTOR alters DEPTOR inhibition, we extensively phosphorylated DEPTOR with

the activated mTORC1 mutant A1459P and assayed mTORC1 inhibition by this pre-phosphorylated DEPTOR. We found almost no inhibition of mTORC1 with pre-phosphorylated DEPTOR (*Figure 7A*). This is consistent with results in cells showing that sustained serum stimulation results in less epitope-tagged, overexpressed wild-type DEPTOR being co-immunoprecipitated with RAPTOR compared with the 13 S/T→A mutant (*Peterson et al., 2009*). Although previous results also have shown that DEPTOR phosphorylation leads to increased ubiquitylation by the F box protein βTrCP and subsequent degradation (*Duan et al., 2011*; *Gao et al., 2011*), our reconstituted system with no degradation machinery, formally establishes that phosphorylation of DEPTOR by mTORC1 results in decreased association of DEPTOR with mTORC1. This supports the previously-proposed mechanism of mTORC1 self-regulation of its inhibition (*Figure 7B*; *Peterson et al., 2009*). It suggests that the degradation that is observed in cells happens subsequent to loss of inhibition of mTORC1 by DEPTOR.

## Discussion

Our structural and functional analysis of reconstituted mTORC1 inhibition by DEPTOR revealed a unique bipartite binding mechanism, involving an interaction of the DEPTOR PDZ with the FAT domain of mTOR and the DEPTOR long-linker interaction with the FRB. We further identified the long-linker-PDZ as a minimal inhibitory unit. However, our kinetic analysis suggests that the tandem DEP domain region also has a role in inhibition, and work by Wälchli et al. concurrently published with ours suggests a mechanism for this (*Wälchli et al., 2021*).

Remarkably, DEPTOR only partially inhibits mTORC1 phosphorylation of its two major substrates 4EBP1 and S6K1, independent of the mTOR activation state, while PRAS40 fully inhibits mTOR activity under the identical assay conditions. Partial inhibition of mTORC1 by DEPTOR in cells has been noted (reviewed in *Caron et al., 2018*). Several previous reports have suggested that DEPTOR dampens mTORC1 activation, rather than fully inhibiting it, leaving a residual mTORC1 specific phosphorylation of S6K1 (*Caron et al., 2016*; *Dong et al., 2017*; *Hu et al., 2017*; *Laplante et al., 2012*; *Li et al., 2014*). However, it has not been established previously that this is due to DEPTOR's unique property of forming an inhibitory complex with mTORC1 that is capable of turning over substrate. This residual mTORC1 activity in the presence of DEPTOR explains why cells inhibited by rapamycin or by DEPTOR overexpression show different phenotypes and why deletion of DEPTOR has a lesser activating effect on mTORC1 in cells than the loss of TCS1/2 (*Caron et al., 2018*). Our results show that even for purified recombinant components, partial inhibition is an intrinsic property of DEPTOR and does not require additional cellular components.

There is evidence that multiple weak interactions may be preferred in biological systems to impart greater specificity (*Scheepers et al., 2020*). Multi-site substrate interactions are a hallmark of mTORC1. Substrates such as 4EBP1 and S6K1 interact with both the mTOR FRB and the RAPTOR TOS-binding site to increase the avidity of mTORC1 for its substrates (*Yang et al., 2017*). Furthermore, simultaneous interaction the RAIP and TOS motifs of 4EBP1 with RAPTOR increase the avidity of the 4EBP1 for mTORC1 and control hierarchical phosphorylation (*Böhm et al., 2021*). Similarly, the two distinct bipartite binding mechanisms of DEPTOR vs. PRAS40 allow a more finely-tuned regulation than single binding sites could provide. DEPTOR, which shares only the FRB-binding site with the substrates, allows for residual mTOR activity, while PRAS40, which shares both the FRB- and TOS-binding sites with the substrates, results in full enzyme inhibition. An additional layer of control is that the partial inhibition arising from the allosteric inhibition by DEPTOR establishes an upper limit on mTORC1 inhibition. This partial inhibition could be key for keeping the balance in the mTOR/PI3K signaling cascade and mimicking this aspect of DEPTOR might be an objective for design and selection of small-molecule inhibitors of mTORC1. Given that the PDZ domain alone is not sufficient for mTORC1 inhibition, it is likely that the unique partial inhibition that we have demonstrated for DEPTOR is due to an additional interaction made by DEPTOR. We have shown that the linker interacts with the FRB, and Wälchli and colleagues have shown that the DEPt also interacts with mTORC1 (*Wälchli et al., 2021*). The role of the PDZ interaction with mTOR is to synergise with these interactions to increase affinity of the DEPTOR for mTORC1. We have shown that the DEPTOR linker interacts with a region on the FRB that partially overlaps with the FRB-binding regions of substrates 4EBP1 and S6K1 as well as the inhibitor PRAS40 (*Yang et al., 2017*). The structure of a peptide from S6K1 bound to the FRB shows a much smaller contact with the FRB than the five helical

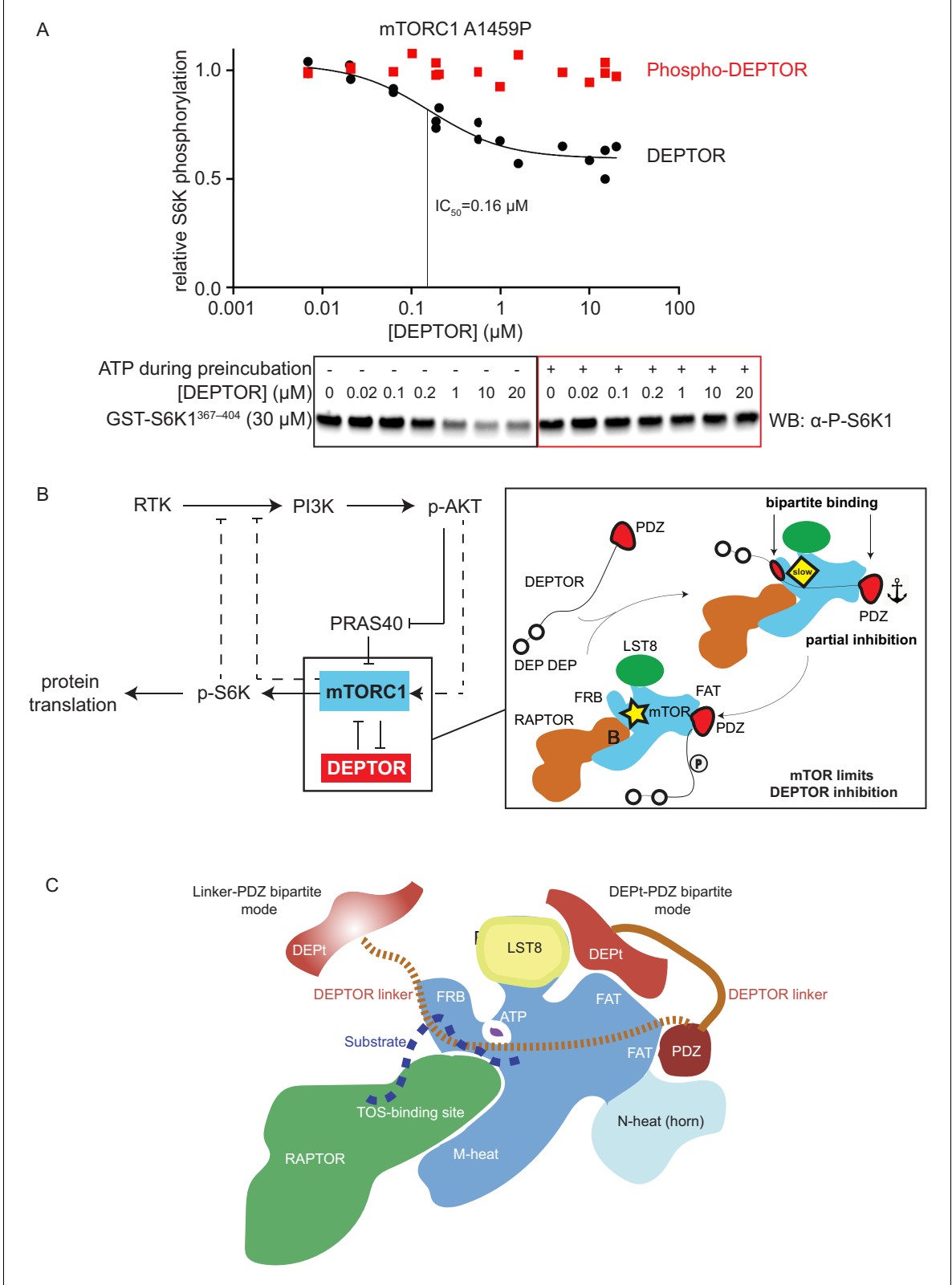

**Figure 7.** mTORC1 directly controls the function of its negative regulators. (**A**) DEPTOR pre-phosphorylated by activated mTORC1 (phospho-DEPTOR) shows no inhibition of mTORC1 activity. Band intensities reflecting P-S6K1$^{367-404}$ peptide were normalized to the control (0 µM DEPTOR). Each data point from three independent experiments is shown with varied [DEPTOR] in each experiment and results were fit by a nonlinear regression to determine IC$_{50}$. (**B**) Negative feedback to PI3K (dashed lines) controls the activity of the two mTORC1 inhibitors PRAS40 and DEPTOR. If mTORC1

*Figure 7 continued on next page*

*Figure 7 continued*

inhibition is high, PI3K is activated due to a loss of the negative feedback. Increased p-AKT results in decreased PRAS40 inhibition, with increased mTORC1 activity, which in turn reduces DEPTOR inhibition. mTORC1 activity alone is sufficient to reduce DEPTOR inhibition, without the necessity of DEPTOR's ubiquitination and subsequent degradation. This increased mTORC1 activity in turn induces negative feedback to PI3K and prevents an overactivation of this pathway, keeping activities balanced. (C) A model for DEPTOR interaction with mTORC1. Taken together, our work and that of *Wälchli et al., 2021* suggest that there are two possible modes of DEPTOR bipartite binding, with both modes having the PDZ domain bound to the PDZ-binding site, while the second interaction involves either: (1) the DEPTOR linker binding to the FRB (our work here) or (2) the DEPt binding to the DEPt binding site (*Wälchli et al., 2021*). The observed inhibition is a sum of the two modes. For the basal mTORC1, bipartite interaction with mTOR via linker and PDZ might predominate. For the RHEB-bound mTORC1, DEPt could play a predominant inhibitory role by preventing allosteric changes necessary for activation. Having two modes of inhibition ensures that the DEPTOR effectively inhibits both mTORC1 (inhibition with both modes) and mTORC2 (presumably only inhibition through the DEPt-PDZ mode), since the FRB site would not be accessible in mTORC2. It is possible that there are hybrid states of the DEPTOR-bound mTORC1. For example, RHEB binding could cause the PDZ binding site to switch to its high-affinity conformation, while the DEPt maintains the active site in its basal conformation.

The online version of this article includes the following source data for figure 7:

**Source data 1.** Uncropped blots.
**Source data 2.** Data values for *Figure 7*.

---

turns from PRAS40 that were observed binding to the FRB. The density that we observe for DEPTOR bound to the FRB is more extensive than the S6K1 interaction, but less extensive (modelled as three helical turns) than what was observed previously for PRAS40. While the structures do not provide an unambiguous mechanism for partial inhibition, these differences in the extent of the interface with the FRB might contribute to DEPTOR being a partial inhibitor and PRAS40 being a full inhibitor. In addition, interactions of the DEPt with mTORC1 also may add to partial inhibition (*Wälchli et al., 2021*).

A major challenge in inhibiting the PI3K/mTOR pathway is the presence of numerous feedback loops that prevent therapeutic success and give rise to resistance against treatment (reviewed in *Yang et al., 2019*). The use of mTOR inhibitors ultimately results in an overactivation of the PI3K/mTOR pathway due to accumulation of IRS1 and IGF-1R triggering a positive feedback loop which diminishes the therapeutic value of treatment (*Britschgi et al., 2012*; *Dibble et al., 2009*; *O'Reilly et al., 2006*). Downregulating mTORC1 activity while maintaining negative feedback loops could be of therapeutic benefit. A more detailed understanding of mTOR function and ways of partially inhibiting some of the mTOR-dependent processes, while maintaining others might therefore be the key to render mTOR a suitable cancer drug target. DEPTOR is highly overexpressed in Multiple Myeloma cells, which protects these cells from apoptosis (*Peterson et al., 2009*). Recently, small molecule inhibitors have been designed to interrupt the DEPTOR PDZ/mTOR interaction and have shown selective cytotoxicity against this type of cancer (*Lee et al., 2017*; *Shi et al., 2016*; *Vega et al., 2019*). Efforts like this could significantly benefit from the structural details of the unique non-canonical binding mode of mTORC1 and DEPTOR PDZ described in this study. The new structural insights are important steppingstones toward selectively targeting the mTOR/DEPTOR interaction by pharmacotherapy. In addition to the unique interface formed by mTOR/DEPTOR PDZ interaction, the short-linker/PDZ interface could serve as a second interface to be targeted by small molecule inhibitors, as this interaction has been uniquely observed in DEPTOR PDZ and stability of the PDZ domain is severely impaired when this interaction is lost. The unique features of the mTORC1/PDZ interaction described in this work could be essential for selectively targeting the PDZ domain, which is one of the most common scaffolding domains in the human proteome with about 270 existing versions present in 150 proteins (*Harris and Lim, 2001*).

The discrepancy between the previously reported observation in cells of reduced DEPTOR binding to activated cancer-associated mutants (*Grabiner et al., 2014*) and our observation of a lowered DEPTOR IC$_{50}$ and increased affinity in our reconstituted system for activated mTORC1 might be explained by upregulated cellular processes in cancer cells. One key observation upon DEPTOR discovery was that DEPTOR inhibition quickly vanished after mTORC1 activation, several hours ahead of DEPTOR degradation (*Duan et al., 2011*; *Gao et al., 2011*; *Peterson et al., 2009*). Besides a reduction in DEPTOR expression levels and an increase in its degradation, there might be other factors that drive DEPTOR dissociation upon mTORC1 activation, such as a change in localization of

proteins (mTORC1 translocates to lysosomes upon mitogen stimulation *Bar-Peled and Sabatini, 2014*), mTORC1 interactors that could outcompete DEPTOR binding (*Yoon et al., 2015*), or a change in the DEPTOR phosphorylation levels (*Duan et al., 2011*; *Gao et al., 2011*; *Peterson et al., 2009*), which could decrease mTORC1 affinity and/or increase DEPTOR affinity for other proteins. Grabiner et al. did not compare DEPTOR binding to mitogen stimulated wild-type mTOR with DEPTOR binding We have shown that a cancer-associated, activating mutation alone appears to increase DEPTOR binding by increasing the affinity of mTORC1 for the DEPTOR PDZ domain (*Figure 6E*). Taking into account the interactions that we see between the DEPTOR linker and the FRB, and the interactions shown between the DEPt and mTOR by *Wälchli et al., 2021*, we propose that there are two binding modes of DEPTOR with mTORC1 (*Figure 7C*). In both modes, the C-terminus of DEPTOR is anchored by the PDZ/mTOR FAT interaction, while the N-terminal region is anchored by the DEPTOR linker interacting with the mTOR FRB in one mode or by the DEPt interaction with the mTOR FAT domain in the other mode (*Wälchli et al., 2021*). Both binding modes contribute to the inhibition of mTORC1 by DEPTOR.

We found that phosphorylated DEPTOR no longer inhibits mTORC1. These findings suggest that DEPTOR regulation is performed by mTORC1 itself (*Figure 7B*). In light of mTORC1 remaining partially active in the presence of DEPTOR, it is clear that the kinase itself can release inhibition by DEPTOR. Future work will be necessary to determine the structural basis for this phosphorylation-dependent release of DEPTOR inhibition and how this release is related to the two types of bipartite interactions that DEPTOR makes with mTORC1. Our results show that an activated mTORC1 binds tighter to DEPTOR, but once it phosphorylates DEPTOR, it releases inhibition by DEPTOR. In contrast, PRAS40 regulation arises from phosphorylation by Akt. Nevertheless, despite the mechanistic differences in DEPTOR-mediated and PRAS40-mediated regulation, the negative feedback loop via p-S6K1 on the PI3K pathway introduces a balance to prevent either too much or too little inhibition by these mTORC1 regulators.

# Materials and methods

## Key resources table

| Reagent type (species) or resource | Designation | Source or reference | Identifiers | Additional information |
|---|---|---|---|---|
| Antibody | Anti-P-4EBP1 (T37/46) (Rabbit, polyclonal) | Cell Signalling | Cat#9459L; RRID:AB_330985 | WB (1:1000) |
| Antibody | Anti-4EBP1 (Rabbit, polyclonal) | Cell Signalling | Cat#9452S; RRID:AB_331692 | WB (1:1000) |
| Antibody | Anti-P-p70 S6 Kinase (T389) (Rabbit, monoclonal) | Cell Signalling | Cat#9205L; RRID:AB_330944 | WB (1:1000) |
| Antibody | Anti-Rabbit IgG, HRP-linked Antibody | Cell Signalling | Cat#7074; RRID:AB_2099233 | WB (1:5000) |
| Strain, strain background (*Escherichia coli*) | LOBSTR cells | KeraFast | Cat# EC1001 | Chemically competent cells |
| Cell line (*Homo sapiens*) | Expi293F cells | Thermo Fisher | Cat#A14527; RRID:CVCL_D615 | |
| Recombinant DNA reagent | pOPL24 (plasmid) | *Anandapadamanaban et al., 2019* | | 2xStrepII-1xFlag-Human_Raptor, in pCAG |
| Recombinant DNA reagent | pOPL25 (plasmid) | *Anandapadamanaban et al., 2019* | | Human_mTOR nontagged, in pCAG |
| Recombinant DNA reagent | pOPL26 (plasmid) | *Anandapadamanaban et al., 2019* | | 3xFlag-Human_LST8, in pCAG |
| Recombinant DNA reagent | pOPL95 (plasmid) | This paper | | Human_mLST8 nontagged, in pCAG (Available from RLW lab) |

*Continued on next page*

*Continued*

| Reagent type (species) or resource | Designation | Source or reference | Identifiers | Additional information |
|---|---|---|---|---|
| Recombinant DNA reagent | pOPL119 (plasmid) | This paper | | Human_Raptor nontagged, in pCAG (Available from RLW lab) |
| Recombinant DNA reagent | pOPL121 (plasmid) | This paper | | 2xStrepII(tev)-Human_mTOR, in pCAG (Available from RLW lab) |
| Recombinant DNA reagent | pOPL146 (plasmid) | This paper | | 2xStrepII(tev)-Human_mTOR_A1459P, in pCAG (Available from RLW lab) |
| Recombinant DNA reagent | pOPL151 (plasmid) | This paper | | 2xStrepII(tev)-Human_mTOR_1376–2549, in pCAG (Available from RLW lab) |
| Recombinant DNA reagent | pOPL107 (plasmid) | This paper | | GST(tev)-Human_mTOR_FRB_2015–2114, in pOPTG (Available from RLW lab) |
| Recombinant DNA reagent | pOPL40 (plasmid) | This paper | | GST(tev)-Human_Rheb1, in pOPTG (Available from RLW lab) |
| Recombinant DNA reagent | pMA9 (plasmid) | *Anandapadamanaban et al., 2019* This paper | | HisLIP(tev)-Human_PRAS40, in pOPTL (Available from RLW lab) |
| Recombinant DNA reagent | pAB87 (plasmid) | This paper | | GST(PreScission)Human_DEPTOR_1–409 (S204, N389 natural variant) (Full length WT), in pGEX-6P1 (Available from RLW lab) |
| Recombinant DNA reagent | pOPL111 (plasmid) | This paper | | GST(tev)-Human_DEPTOR_228_409 (Linker-PDZ), in pOPTG (Available from RLW lab) |
| Recombinant DNA reagent | pOPL127 (plasmid) | This paper | | HisLIP(tev)-Human_DEPTOR_324_409 (PDZ), in pOPTL (Available from RLW lab) |
| Recombinant DNA reagent | pOPL138 (plasmid) | This paper | | GST(tev)-Human_DEPTOR_1–409 (S204, N389 natural variant) (Full length WT), in pOPTG (Available from RLW lab) |
| Recombinant DNA reagent | pOPL139 (plasmid) | This paper | | GST(tev)-Human_DEPTOR_1-409-13S/T-A mutant (T241A,S244A,T259A, S260A,S263A, S265A,S282A,S283A, S287A, S293A,S297A, S298A,S299A) (Full length 13A), in pOPTG (PCR-ed from Addgene clone 21702) (Available from RLW lab) |
| Recombinant DNA reagent | pOPL140 (plasmid) | This paper | | GST(tev)-Human_DEPTOR_1–323 (DEP-DEP-linker), in pOPTG (Available from RLW lab) |
| Recombinant DNA reagent | pOPL141 (plasmid) | This paper | | GST(tev)-Human_DEPTOR_1-323-13S/T-A mutant (DEP-DEP-linker 13A), in pOPTG (Available from RLW lab) |

*Continued on next page*

*Continued*

| Reagent type (species) or resource | Designation | Source or reference | Identifiers | Additional information |
|---|---|---|---|---|
| Recombinant DNA reagent | pOPL143 (plasmid) | This paper | | GST(tev)-Human_DEPTOR_228-409-13S/T-A mutant (linker PDZ 13A), in pOPTG (Available from RLW lab) |
| Recombinant DNA reagent | pOPL157 (plasmid) | This paper | | GST(tev)-Human_DEPTOR_1–220 (DEP-DEP), in pOPTG (Available from RLW lab) |
| Recombinant DNA reagent | pOPL159 (plasmid) | This paper | | His(tev)-Human_DEPTOR_228–323 (Linker), in pOPTH(tev) (Available from RLW lab) |
| Recombinant DNA reagent | pOPL166 (plasmid) | This paper | | GST(tev)-Human_DEPTOR_305–409 (extended PDZ), in pOPTG (Available from RLW lab) |
| Recombinant DNA reagent | pOPL167 (plasmid) | This paper | | GST(tev)-Human DEPTOR_315–409 (extended PDZ), in pOPTG (Available from RLW lab) |
| Recombinant DNA reagent | pOPL163 (plasmid) | This paper | | GST(PreScission)-Human_S6K1αII, 367–404, in pGEX-6P1 (Available from RLW lab) |
| Recombinant DNA reagent | pOP826 (plasmid) | This paper | | in GST(tev)-Human 4EBP1, in pOPTG (Available from RLW lab) |
| Recombinant DNA reagent | pOP854 (plasmid) | This paper | | GST(tev)-Human 4EBP1-TOS-less-(114-FEMDI-118/AAAAA), in pOPTG (Available from RLW lab) |
| Chemical compound, drug | NEBuilder HiFi assembly Master Mix | New England Biolabs | Cat#E2621S | |
| Chemical compound, drug | cOmplete EDTA-free protease inhibitor tablets | Roche | Cat#11873580001 | |
| Chemical compound, drug | Universal Nuclease | Pierce | Cat#88702 | |
| Chemical compound, drug | Desthiobiotin | IBA | Cat#2-1000-005 | |
| Chemical compound, drug | Glutathione Sepharose 4B resin | GE Healthcare | Cat#17-0756-05 | |
| Chemical compound, drug | Ni-NTA agarose resin | Qiagen | Cat#30230 | |
| Chemical compound, drug | Lysozyme | Sigma | Cat#L6876 | |
| Chemical compound, drug | TCEP (Tris(2-carboxyethyl) phosphine hydrochlorid) | Soltec Ventures | Cat#M115 | |
| Chemical compound, drug | Bovine Serum Albumin | Thermo Fisher | Cat#BP1605 | |
| Chemical compound, drug | PEI (Polyethyleneimine 'MAX', MW 40,000) | Polysciences | Cat#24765 | |

*Continued on next page*

Continued

| Reagent type (species) or resource | Designation | Source or reference | Identifiers | Additional information |
|---|---|---|---|---|
| Chemical compound, drug | SuperSignal West Pico PLUS chemiluminescent substrate | Thermo Fisher | Cat#34577 | |
| Chemical compound, drug | LDS sample buffer | NuPAGE | Cat#NP0008 | |
| Chemical compound, drug | Triton X-100 | Sigma | Cat#T8787 | |
| Chemical compound, drug | Tween 20 | NBS Biologicals | Cat#17767-B | |
| Chemical compound, drug | ATP | Jena Bioscience | Cat#NU-1010 | |
| Chemical compound, drug | GTPγS | Jena Bioscience | Cat#NU-412–20 | |
| Chemical compound, drug | Glutaraldehyde solution, 25% in water | Sigma | Cat#G5882 | |
| Chemical compound, drug | $D_2O$ | Acros Organics | Cat#351430075 | |
| Chemical compound, drug | InstantBlue Protein Stain | Expedeon | Cat#1SB1L | |
| Chemical compound, drug | Expi293 Expression Medium | Thermo Fisher | Cat#A1435102 | |
| Chemical compound, drug | Yeast Nitrogen Base Without Amino Acids and Ammonium Sulfate | Sigma | Cat#Y1251 | |
| Chemical compound, drug | Ammonium 15N chloride | Sigma | Cat#299251 | |
| Chemical compound, drug | D-Glucose (U-$^{13}C_6$ 99%) | Cambridge Isotopes Laboratores Inc | Cat#CLM-1396–10 | |
| Chemical compound, drug | 99.9% 2H atom $D_2O$ | Cortecnet | Cat#CD5251P1000 | |
| Software, algorithm | ASTRA software package for analysis of SEC-MALS data | Wyatt | http://www.wyatt.com/products/software/astra.html; RRID:SCR_016255 | |
| Software, algorithm | ProteinLynx Global Server | Waters | 720001408EN; RRID:SCR_016664 | |
| Software, algorithm | DynamX | Waters | 720005145EN | |
| Software, algorithm | ImageJ | *Schneider et al., 2012* | https://imagej.nih.gov/ij/; RRID:SCR_003070 | |
| Software, algorithm | GraphPad Prism | GraphPad Software | https://www.graphpad.com/ RRID:SCR_002798 | |
| Software, algorithm | Topspin | Bruker | https://www.bruker.com/products/mr/nmr/software/topspin.html; RRID:SCR_014227 | |
| Software, algorithm | NMRFAM-Sparky | *Lee et al., 2015* | http://pine.nmrfam.wisc.edu/download_packages.html RRID:SCR_014228 | |
| Software, algorithm | MARS | *Jung and Zweckstetter, 2004* | https://www3.mpibpc.mpg.de/groups/zweckstetter/_links/software_mars.html | |
| Software, algorithm | NMRPipe | *Delaglio et al., 1995* | https://www.ibbr.umd.edu/nmrpipe/install.html | |

*Continued*

| Reagent type (species) or resource | Designation | Source or reference | Identifiers | Additional information |
|---|---|---|---|---|
| Software, algorithm | MDD compressed sensing | *Kazimierczuk and Orekhov, 2011* | http://mddnmr.spektrino.com/ | |
| Software, algorithm | POMONA | *Shen and Bax, 2015* | https://spin.niddk.nih.gov/bax/nmrserver/pomona/ | |
| Software, algorithm | Rosetta and NMR restraints | *Raman et al., 2010* | https://www.rosettacommons.org/software | |
| Software, algorithm | Random coil shifts | *Kjaergaard and Poulsen, 2011*; *Kjaergaard and Poulsen, 2011*; *Schwarzinger et al., 2001*. | https://www1.bio.ku.dk/english/research/bms/sbinlab/randomchemicalshifts1/ | |
| Software, algorithm | RELION three software package | *Scheres, 2012* | http://www2.mrc-lmb.cam.ac.uk/relion; RRID:SCR_016274 | |
| Software, algorithm | Gctf | *Zhang, 2016* | RELION three software package | |
| Software, algorithm | MotionCor2 | *Zheng et al., 2017* | RELION three software package | |
| Software, algorithm | ResMap | *Kucukelbir et al., 2014* | RELION three software package | |
| Software, algorithm | Chimera | *Pettersen et al., 2004* | http://plato.cgl.ucsf.edu/chimera/ RRID:SCR_004097 | |
| Software, algorithm | Coot | *Casañal et al., 2020*; *Emsley and Cowtan, 2004* | https://www2.mrc-lmb.cam.ac.uk/personal/pemsley/coot/; RRID:SCR_014222 | |
| Software, algorithm | ChimeraX | *Pettersen et al., 2021* | https://www.cgl.ucsf.edu/chimerax/ RRID:SCR_015872 | |
| Software, algorithm | REFMAC5 | *Brown et al., 2015*; *Murshudov et al., 1997* | https://www.ccp4.ac.uk; RRID:SCR_007255 | |
| Other | 0.45 µM Syringe Filter | Millipore Sigma | Cat#SE2M230I04 | |
| Other | 5 µM Minisart Syringe Filter | Sartorius | Cat#17594-Q | |
| Other | 5 mL StrepTrap HP column | GE Healthcare | Cat#28-9075-47 | |
| Other | 5 mL HiTrap Q HP column | GE Healthcare | Cat#17-1153-01 | |
| Other | 5 mL HiTrap Heparin HP column | GE Healthcare | Cat#17-0407-01 | |
| Other | 5 mL HisTrap FF column | GE Healthcare | Cat#17-5255-01 | |
| Other | HiLoad 16/60 Superdex 200 column | GE Healthcare | Cat#17-1069-01 | |
| Other | HiLoad 16/60 Superdex 75 column | GE Healthcare | Cat#17-1068-01 | |
| Other | Superose 6 increase 10/300 GL | GE Healthcare | Cat#29-0915-96 | |
| Other | 4–12% BisTris NuPAGE Protein Gel | Thermo Fisher | Cat#NP0323 | |
| Other | SuperSep Phos-tag 7.5% precast gels | FUJIFILM Wako Chemicals | Cat#192–17381 | |

*Continued on next page*

*Continued*

| Reagent type (species) or resource | Designation | Source or reference | Identifiers | Additional information |
|---|---|---|---|---|
| Other | iBlot 0.2 µM pore size nitrocellulose Transfer Stacks | Thermo Fisher | Cat#IB301002 | |
| Other | 15 mL Amicon Ultra-15 3K Centrifugal Filters | Millipore Sigma | Cat#UFC900324 | |
| Other | 15 mL Amicon Ultra-15 30K Centrifugal Filters | Millipore Sigma | Cat#UFC903024 | |
| Other | 15 mL Amicon Ultra-15 100K Centrifugal Filters | Millipore Sigma | Cat#UFC910024 | |
| Other | 4 mL Amicon Ultra-4 100K Centrifugal Filters | Millipore Sigma | Cat#UFC810024 | |
| Other | Enzymate Pepsin Column | Waters | Cat#186007233 | |
| Other | Acquity UPLC BEH C18 VanGuard Pre-column | Waters | Cat#186003975 | |
| Other | Acquity UPLC BEH C18 column | Waters | Cat#186002346 | |

## Resource availability

Further information and requests for resources and reagents should be directed to and will be fulfilled by the Lead Contact, Roger L. Williams (rlw@mrc-lmb.cam.ac.uk).

The cryo-EM map and the model are deposited with the EMDB (wild-type mTORC1/DEPTOR complex is entry EMD-13099 and A1459P-mTORC1/DEPTOR complex is entry EMD-13097) and PDB (wild-type mTORC1/DEPTOR complex is entry 7OX0 and A1459P-mTORC1/DEPTOR complex entry 7OWG). Backbone assignments of DEPTOR PDZ, mTOR-FRB and DEPTOR linker have been submitted to the BMRB, Biological Magnetic Resonance Bank with the accession numbers 50324, 50325 and 50326, respectively.

## Experimental model and subject details

*E. coli* C41(DE3)-RIPL cells were used for expression of all DEPTOR constructs as well as for 4EBP1, S6K1, RHEB and the isolated mTOR FRB domain. *E. coli* LOBSTR cells (EC1001, KeraFast) were used for the expression of PRAS40. For unlabeled proteins, the cells were grown in 2xTY media at 37°C, induced at the $OD_{600}$ = 0.7 with 0.3 mM isopropyl-d-1-thiogalactopyranoside, followed by 16 hr of growth at 18°C before harvest. To express proteins labeled for NMR studies, cells were grown in supplemented minimal media, as described in the 'NMR sample preparation' section of the detailed methods. Expi293F cells (Thermo Fisher A14527, RRID:CVCL_D615) were used for the production of mTORC1 and its mutants. Cells were grown in a Multitron Pro shaker set at 37°C, 8% $CO_2$ and 125 rpm. Cells were transfected at a cell density of $2.5 \times 10^6$ cells/mL by co-transfecting plasmids (1.1 mg total DNA/L cells) using PEI (Polyethyleneimine 'MAX', MW 40,000, Polysciences, 24765, total 3 mg PEI/L cells).

## Method detail

### Recombinant protein expression and purification

For DEPTOR (WT), DEPTOR 13S/T-A, DEPDEP, DEPDEP linker and linker-PDZ, a cell pellet of a 12 L *E. coli* culture was resuspended in 75 mL of lysis buffer (50 mM Tris pH 8, 100 mM NaCl, 1 mM TCEP). A description of the growth of constructs labeled for NMR can be found in the 'NMR sample preparation' section below. Two Complete EDTA-free inhibitor tablets (Roche), 500 µL of a 100 mM PEFA solution and 40 mg of lysozyme were added to the cell suspension, which was subsequently sonicated and centrifuged at 35,000 rpm for 35 min. The resulting supernatant was incubated with 3 mL equilibrated Glutathione-Sepharose 4B beads (GE Healthcare) for 45 min at 4°C, while rolling at 18 rpm. Sedimentation of the beads for 2 min at 600 g and the careful removal of the unbound fraction was followed by extensive washing with lysis buffer under gravity flow conditions. Bound

DEPTOR was incubated with TEV protease overnight at 4°C. Cleaved DEPTOR protein was then collected and applied to a 5 mL HiTrap Heparin HP column (GE Healthcare) equilibrated with HEP-A buffer (50 mM HEPES pH 8, 100 mM NaCl, 1 mM TCEP). After washing the column with 80 mL of HEP-A buffer, DEPTOR was eluted with a gradient using HEP-B buffer (50 mM HEPES pH 8, 450 mM NaCl, 1 mM TCEP). DEPDEP was purified via a 5 mL HiTrap Q column (GE Healthcare) in place of the Heparin column using identical buffer composition. A final size-exclusion chromatography step on a S75 16/60 column was performed (50 mM HEPES pH 8, 100 mM NaCl, 1 mM TCEP) and fractions were analyzed by SDS-PAGE. DEPTOR-containing fractions were combined and concentrated to 23 mg/mL. The purified DEPTOR protein was flash frozen in liquid nitrogen and stored at −80°C.

After cell lysis as described for DEPTOR (WT), the His-lipoyl-tagged DEPTOR PDZ was loaded onto a 5 mL NiNTA column and washed with 75 mL lysis buffer containing 10 mM imidazole prior to eluting the protein with 25 mL lysis buffer spiked with 300 mM imidazole. TEV cleavage was performed overnight as described above. The buffer salt concentration was diluted to 30 mM NaCl and the solution was flown through a 5 mL HiTrap Q HP (GE Healthcare) and a NiNTA column. The flow-through was collected and concentrated before gel filtration as described for DEPTOR (WT). mTORC1 complexes were expressed by transient transfection of Expi293F cells grown in Expi293 media. A total of 1.1 mg DNA/L cells was co-transfected into cells at a density of $2.5 \times 10^6$ cells/mL using PEI (Polyethyleneimine 'MAX', MW 40,000, Polysciences, 24765, total 3 mg PEI/L cells). After 48 hr, cells were harvested by centrifugation and cell pellets were frozen in liquid $N_2$. Cell pellet from 2 L culture was resuspended in 200 mL of the lysis buffer (50 mM Tris-HCl, pH 8, 500 mM NaCl, 10% glycerol, 1 mM TCEP, 1 mM EDTA, 1 mM EGTA), supplemented with six Complete EDTA-free protease-inhibitor tablets (Roche), 5 µL Universal nuclease (Pierce, 250 U/µL), and 400 µL of a 100 mM PEFA solution, using a Dounce homogenizer (Kontes, 100 mL, Pestle B, small clearance) on ice and sonication (2x 15 s ON at 40% amplitude). The cell lysate was spun at 15,000 rpm for 35 min in a Ti45 rotor, then filtered through Minisart 5 µm filter. Two tandem Strep-Trap HP columns (GE Healthcare 28-9075-48) were equilibrated with lysis buffer, then the filtered lysate was loaded onto the column at a flow of 2.5 mL/min. Extensive washes with lysis buffer (>200 mL), with 50 mL lysis buffer supplemented with 200 mM $Li_2SO_4$, and 50 mL of the TEV cleavage buffer (50 mM HEPES pH 7.5, 150 mM NaCl, 10% glycerol, 1 mM TCEP) were performed prior to loading 0.1 mg/mL TEV protease onto the column. The cleavage reaction was performed on the column overnight. The protein was then eluted and the salt concentration was adjusted to 50 mM NaCl. The protein was then loaded onto a 5 mL HiTrap Q column (GE Healthcare) equilibrated with 50 mM HEPES pH 7.5, 50 mM NaCl, and was eluted via a salt gradient. mTORC1-containing fractions were concentrated using an Amicon Ultra-4 100 kDa concentrator, spinning in 3 min intervals at 1000 rcf. Gel filtration was performed using a Superose 6 Increase (10/300) column in gel filtration buffer (50 mM HEPES pH 7.5, 200 mM NaCl, 1 mM TCEP). mTOR complex fractions were pooled and again concentrated using an Amicon Ultra-4 100 kDa concentrator before freezing the protein.

Human RHEB was cloned into a pOPTG vector encoding an N-terminal GST-tag followed by a TEV cleavage site for proteolytic removal. The fusion protein was overexpressed in *E. coli* C41(DE3) RIPL cells as described for DEPTOR. Cells were lysed in 30 mM Tris pH 8.0, 200 mM NaCl, 0.3 mM TCEP, 1.5 mM MgCl$_2$ and 5% glycerol, and GST-fused RHEB was purified by an affinity chromatography step on Glutathione-Sepharose 4B beads. The GST-tag was removed by incubating with TEV protease overnight. To separate the HIS6-TEV protease from the RHEB protein, a nickel-affinity chromatography step was performed before a final gel filtration step on a Superdex 75 16/60 column with 30 mM Tris pH 8.0 (4 °C), 100 mM NaCl, 5 mM MgCl$_2$ and 1 mM TCEP, which yielded purified RHEB. Removal of bound GDP was achieved by incubating the protein for 1 hr on ice with EDTA buffer containing 20 mM HEPES pH 7, 100 mM NaCl, 20 mM EDTA and 1 mM TCEP. Next, the buffer was exchanged to 50 mM HEPES pH 7, 100 mM NaCl, 5 mM MgCl$_2$, 1 mM TCEP using an Amicon Ultra-4 4 kDa concentrator before concentrating the protein to 15.5 mg/mL. Purified RHEB was flash frozen in liquid nitrogen and stored at −80°C.

Human PRAS40 was purified as described previously (*Anandapadamanaban et al., 2019*).

## mTORC1 activity assays

All reactions were performed in kinase buffer (KB) consisting of 25 mM HEPES pH 7.4, 75 mM NaCl, 0.9 mM TCEP, 5% glycerol, at 30°C for a duration of 45 min for non-activated mTORC1, 2 to 4 min

for activated mTORC1, and at 20°C for 20 min for the unstable DEPTOR PDZ construct using non-activated mTORC1. Reactions were set up by preincubating 100 nM non-activated mTORC1 with either 5 µM 4EBP1, 10 µM TOS-less 4EBP1 or 30 µM GST-S6K1$^{367-404}$ peptide as substrates and various concentrations of the inhibitors for 10 min on ice. After 30 s temperature equilibration at 30°C, reactions were started by the addition of 75 µM ATP and 10 mM MgCl$_2$. For the activated mTORC1 complexes (mTOR-A1459P mutant or in the presence of RHEB-GTP), 30 µM 4EBP1 or GST-S6K1$^{367-404}$ peptide and 20 nM mTORC1 were used and the reaction was started with 500 µM ATP and 10 mM MgCl$_2$. For RHEB-activated mTORC1, RHEB was preincubated for 1 hr with a 30-fold molar excess of GTPγS (Jena Bioscience NU-412–20, lot IT008-18). Next, mTORC1, DEPTOR, 4EBP1 and RHEB-GTPγS were mixed in the order of mentioning and preincubated for 10 min on ice. After 30 s temperature equilibration at 30°C, reactions were started by the addition of 500 µM ATP and 10 mM MgCl$_2$. All reactions were quenched with 2x SDS sample buffer and resolved on a 4–12% NuPage Bis-Tris gel. Western blots were performed using a 0.2 µM pore size nitrocellulose membrane (Invitrogen IB301002) and the iBlot dry blotting transfer system (Invitrogen). Blocking was performed using 5% Marvel in TBST buffer (100 mM Tris-HCl, 150 mM NaCl, 0.1% Tween 20). Antibodies were obtained from Cell Signalling (P-4EBP1 (T37/46) Rabbit AB, 9459L, RRID:AB_330985; 4EBP1 Rabbit AB, 9452S, RRID:AB_331692; P-p70 S6 Kinase (T389) Rabbit, 9205L, RRID:AB_330944; Anti-Rabbit IgG, HRP-linked Antibody, 7074, RRID:AB_2099233). Antibody solutions contained 5% BSA in TBST, using 1:1000 dilution for primary antibodies and 1:5000 for the secondary antibody. Detection was performed using the Bio-Rad ChemiDoc-Touch Imaging System.

For the inhibition of mTORC1 A1459P mutant by DEPTOR measured by Phos-tag SDS PAGE, the reactions containing 30 nM mTORC1, 15 µM 4EBP1, 250 mM ATP, 10 mM MgCl$_2$ and varied concentration of DEPTOR were incubated for 4 min at 30°C and then quenched by the addition of 2X SDS sample buffer. The samples were analyzed by SuperSep Phos-tag (50 µM), 7.5% precast gels (192–17381), with MOPS –EDTA running buffer supplemented with 5 mM sodium bisulphate. Staining was done using InstantBlue and quantification followed using the Bio-Rad ChemiDoc-Touch Imaging System.

To analyze phosphorylation of DEPTOR by mTORC1, assays contained 30 nM mTORC1-A1459P mutant, 250 µM ATP, 10 mM MgCl$_2$ and 20 µM of each DEPTOR variant. Phosphorylation was detected by SuperSep Phos-tag precast gels using InstantBlue stain and Bio-Rad ChemiDoc-Touch Imaging System.

To compare the inhibition by DEPTOR vs. phospho-DEPTOR, reactions were set up using a dilution series of DEPTOR (30–0 µM) and 60 nM mTORC1-A1459P mutant. Reaction mix was split in two, one reaction was started by adding 20 mM MgCl$_2$ and 500 µM ATP (to produce phospho-DEPTOR), the second mix was spiked with an equal volume of KB (DEPTOR as a control). After 2 hr, an aliquot of each reaction was quenched in SDS buffer and analyzed by Phos-tag gel for the completeness of mTORC1 dependent DEPTOR phosphorylation ($\geq$80%). At the same time, a second aliquot of each reaction was added into substrate mix resulting in a final concentration of 30 µM GST-S6K, 500 µM ATP, and 20 mM MgCl$_2$ in KB to test for inhibition of phospho-DEPTOR vs. DEPTOR. The reaction was quenched after 2 min, and the results were analyzed as described above by western blots.

All IC$_{50}$ were determined by the non-linear regression $y=y_{min} + (y_{max}-y_{min})/(1+([DEPTOR]/IC_{50}))$. $K_{M,4EBP1}$ in the presence of DEPTOR was fit according to the Michaelis Menten equation $y = v_{max}*[4EBP1]/(K_M + [4EBP1])$. Statistical analysis and curve fitting for IC$_{50}$ experiments was performed in GraphPad Prism (RRID:SCR_002798). Each individual run was normalized to [inhibitor] = 0 µM. Standard deviation of at least three independent replicates is shown for each data point in the graph. Standard deviation for the fit IC$_{50}$, for the residual activity ($y_{min}$) and for $K_{M,4EBP1}$ is reported in the figures.

## Cryo-EM sample preparation

Purified wild-type mTORC1 (0.7 µM) and DEPTOR (1.4 µM) were mixed in ~300 µL and incubated for 30 min on ice. Following the addition of 0.02% glutaraldehyde (Sigma G5882) from a 1% glutaraldehyde stock in GraFix buffer A (50 mM HEPES pH 7.5, 0.1 M NaCl, 1 mM TCEP, 10% glycerol), the sample was immediately subjected to a gradient fixation (GraFix) (*Kastner et al., 2008*). For that, the sample was loaded on a 12 mL gradient of 10–30% glycerol and 0–0.2% glutaraldehyde in 50

mM HEPES pH 7.5, 0.1 M NaCl, 1 mM TCEP performed in a SW40 rotor tube (Ultra-Clear, Beckman 344060) using a gradient maker (Biocomp Instruments). The sample was centrifuged in a SW40 rotor (Beckman) at 33,000 rpm for 14 hr. After centrifugation, 0.45 mL fractions were collected manually from top of the tube, analyzed by SDS-PAGE, and the fractions containing crosslinked material were pooled, quenched by the addition of final 50 mM Tris pH 7.5 and concentrated to 250 μL using an Amicon Ultra-15 100 kDa concentrator. Cross-linked mTORC1/DEPTOR was further run on a Superose 6i 10/300 column equilibrated in 50 mM HEPES pH7.5, 200 mM NaCl, 0.3% CHAPS and 1 mM TCEP and the peak fractions were concentrated to $OD_{280}$ = 0.11 used immediately for cryo-EM grid preparation.

Purified mutant mTORC1 A1459P (1.2 μM) and DEPTOR (6.8 μM) were preincubated in the presence of 1 mM $MgCl_2$ and 500 μM AMP-PNP (Jena Biosciences) for 20 min and used for cryo-EM grid preparation.

## Cryo-EM data collection and processing

Holey carbon Quantifoil Au R 1.2/1.3 (300 mesh) grids were glow-discharged using an Edwards Sputter Coater S150B for 60 s at 40 mA. The grids were covered with graphene oxide as previously described (*Boland et al., 2017*). In short, 3 μL of a 0.2 mg/mL graphene oxide dispersion (Sigma, cat number 777676) was added to the carbon side of the grids and incubated for 1 min. Excess of graphene oxide solution was removed by blotting with Whatman No.1 filter paper and washed three times with Milli-Q $H_2O$ before air-drying for 5 min at room temperature. Graphene oxide covered grids were stored in a grid box overnight.

A 3 μL aliquot of freshly-prepared crosslinked mTORC1/DEPTOR complex at $OD_{280}$ = 0.11 was added to graphene oxide-covered grids and blotted for 11–13 s at 4°C and then plunge-frozen in liquid ethane using a custom-fabricated manual plunger (MRC Laboratory of Molecular Biology). A total of 2370 micrographs of the human mTORC1/DEPTOR complex were acquired on a FEI Titan Krios electron microscope operated at 300 keV. Zero-energy loss images were recorded on a Gatan K2 Summit direct electron detector operated in super-resolution mode with a Gatan GIF Quantum energy filter (20 eV slit width) using SerialEM (*Mastronarde, 2005*) for automated collection. Images were recorded at a calibrated magnification of 34,965 (pixel size of ~1.43 Å) with a dose rate of ~2.5 electrons/$Å^2$/s. An exposure time of 16 s was fractionated into 20 movie frames adding to a total dose of 40 electrons/$Å^2$. For data collection, a defocus-range was set to −1.6 to −3.2 μm (*Table 1*).

A 3 μL aliquot of mutant mTORC1 A1459P/DEPTOR complex was added onto UltrAuFoil R 1.2/1.3 Au 300 mesh (Quantifoil Micro Tools GmbH) after the grid was glow-discharged using an Edwards Sputter Coater S150B for 60 s at 40 mA. Plunge freezing was performed using Vitrobot (Thermo Fisher Scientific) with a blotting time of 2 s at 14°C and 95% humidity and a force of −15. A total of 4759 micrographs of the mutant mTORC1 A1459P /DEPTOR complex were acquired on a FEI Titan Krios electron microscope operated at 300 keV. Zero-energy loss images were recorded on a Gatan K3 Summit direct electron detector operated in super-resolution mode with a Gatan GIF Quantum energy filter (20 eV slit width) using EPU (Thermo Fisher Scientific) for automated collection. Images were recorded at a calibrated magnification of 81,000 (pixel size of ~1.1 Å) with a dose rate of ~1.12 electrons/$Å^2$/s. An exposure time of 3.25 s was fractionated into 50 movie frames adding to a total dose of 56 electrons/$Å^2$. For data collection, a defocus-range was set to −1.4 to −3.0 μm.

## Image processing

All image-processing steps were done using the RELION three software package (*Scheres, 2012*, RRID:SCR_016274), which includes Gctf (*Zhang, 2016*), MotionCor2 (*Zheng et al., 2017*), and ResMap (*Kucukelbir et al., 2014*). Micrographs were processed using GPU-accelerated MotionCor2 to correct for electron beam-induced sample motion, while contrast transfer function (CTF) parameters were determined using Gctf. Reference-based autopicking was performed on the full dataset using Relion3 with initial templates obtained from a previous mTORC1/DEPTOR dataset.

For the wild-type mTORC1/ DEPTOR data set, 491,404 particles were extracted with a particle box size of 400 by 400 pixels. Two rounds of reference-free 2D classification (using a mask with a diameter of 350 Å) resulted in a selection of 390,636 particles. This set of particles was subjected to a 3D classification over 30 iterations in point group $C_1$ using a low-pass filtered (50 Å) ab-initio

**Table 1.** Cryo-EM data collection and processing of mTORC1 in complex with DEPTOR.

**Data collection**

| Protein details | mTORC1 WT/DEPTOR | mTORC1 A1459P/DEPTOR |
|---|---|---|
| Microscope | Titan Krios (FEI) | Titan Krios (FEI) |
| Voltage (kV) | 300 | 300 |
| Detector | Gatan K2 Summit | Gatan K3 |
| Pixel size (Å) | 1.43 | 1.1 |
| Defocus range (μm) | −1.6 to −3.2 | −1.4 to −3.0 |
| Movies | 2370 | 4759 |
| Frames/movie | 20 | 50 |
| Exposure rate ($e^-$ / $Å^2$/ s) | 2.5 | 22.4 |
| Total dose ($e^-$ / $Å^2$) | 40 | 56 |
| Number of particles | 491,404 | 97,314 |
| Energy filter slit width (eV) | 20 | 20 |
| Model composition | | |
| Non-hydrogen atoms | 28917 (refined as a monomer) | 28784 (refined as dimer) |
| Protein residues | 3640 | 7196 |
| Ligands/ ions | - | - |
| Density refinement | | |
| Resolution (Å) | 4.2 | 4.7 |
| Sharpening B-factor (Å) | 283.5 | 145.7 |
| Model refinement | | |
| Root-mean-square deviation | | |
| Bond lengths (Å) | 0.0131 | 0.0089 |
| Bond angles (°) | 1.73 | 1.76 |
| *Validation* | | |
| Molprobity score | 1.47 | 1.4 |
| Clashscore, all atoms | 0.95 | 1.3 |
| Favored rotamers (%) | 98.0 | 89.4 |
| Ramachandran plot (%) | | |
| Favored | 92.9 | 92.2 |
| Allowed | 6.4 | 7.6 |
| Outliers | 0.75 | 0.2 |

reference which was created using the SGD algorithm for de novo 3D model generation introduced in Relion3. Selection of reasonably looking classes by visualization in Chimera (RRID:SCR_004097) and by paying attention to the rotational and translational accuracies for six classes reduced the number of particles to 333,462 sorted into five 3D classes. Without providing a mask around the mTORC1/DEPTOR complex, the subsequent 3D auto-refinement of these particles applying C2 symmetry led to a reconstruction of 6.7 Å resolution, based on the gold-standard FSC = 0.143 criterion (*Rosenthal and Henderson, 2003*; *Scheres, 2012*).

To correct for beam-induced particle movements, increase the signal-to-noise ratio for all particles, and to apply radiation-damage weighting, the refined particles were further 'polished' using the Bayesian approach implemented in Relion3. Following this step, another 3D autorefinement in C2 using a mask around the mTORC1/DEPTOR complex as well as applying solvent-flattened FSCs yielded a reconstruction of 4.3 Å resolution (FSC = 0.143 criterion). After a CTF- and beamtilt-refinement for the estimation of per-particle defoci and beamtilt values for the complete set of selected particles, a subsequently performed 3D autorefinement resulted in a mTORC1/DEPTOR

reconstruction of 4.2 Å resolution (FSC = 0.143 criterion). A similar processing strategy was performed for the mutant mTORC1 A1459P/DEPTOR complex data set, after automated micrograph assessment using MicAssess (*Li et al., 2020*). A total of 97,314 particles were used for refinement, resulting in a reconstruction of 4.7 Å resolution.

To improve resolution (especially in the DEPTOR region), we expanded the wild-type mTORC1/DEPTOR dataset using the relion_particle_symmetry expand command while applying C2 symmetry and performed focussed classfication with signal subtraction (*Bai et al., 2015*). This strategy yielded higher resolution in previous mTORC1 structures (*Anandapadamanaban et al., 2019*; *Yang et al., 2017*). Using focussed 3D-classification with signal subtraction on the expanded 666,924 monomer particles without image alignment yielded two reasonable classes with a total of 464,013 monomers. A 3D refinement of this set of particles corresponding to monomer density led to a reconstruction of 4.4 Å, based on the gold-standard FSC = 0.143 criterion.

To improve the density map of DEPTOR PDZ region, we performed another focused classification with signal subtraction on the monomer particles. A mask was applied to the region of interest on the PDZ (DEPTOR residues 324–409) and surrounding mTOR domains (N-heat and FAT domain, 61–903 and 1474–1644, respectively), particles were 3D classified without image alignment, and the best class was selected for further refinement of the original (unmasked) particles. This resulted in smaller subsets of the original 223,576 particles; in which the PDZ density was better defined. A 3D refinement of the above selected particles resulted in a map at an overall 4.3 Å resolution, based on the gold-standard FSC = 0.143 criterion (*Figure 3—figure supplement 2H*).

## Cryo-EM model building and refinement

After correction for the detector modulation transfer function (MTF) and B-factor sharpening, the post-processed map was used for inspection in Chimera (*Pettersen et al., 2004*) and model building in Coot (*Casañal et al., 2020*; *Emsley and Cowtan, 2004*). Superimposing basal-state monomer and RHEB-activated monomer models of mTORC1 taken from previously reported structures (PDB 6BCX and 6BCU, respectively), it was apparent that the 6BCX agreed better with the mTORC1/DEPTOR density. The 6BCX model was broken into domains that were rigidly fit with COOT to the observed density. The model was manually fit to the density using ISOLDE (*Croll, 2018*) in ChimeraX (*Pettersen et al., 2021*) (RRID:SCR_015872). The CS-Rosetta model for the DEPTOR PDZ domain was manually fit to the density. Backbone dihedral angles were predicted for each residue using Talos-N (*Shen and Bax, 2013*). These were used as external torsion angle restraints for real-space refinement in COOT (RRID:SCR_014222) and for REFMAC5 (*Brown et al., 2015*; *Murshudov et al., 1997*) (RRID:SCR_007255). The model was refined in REFMAC with external (PDB 6BCX) restraints for mTORC1 from PROSMART (*Kovalevskiy et al., 2016*; *Nicholls et al., 2017*) and self, H-bond and Talos-N restraints for the PDZ domain. Manual building and refinement were iterated. A symmetrized dimer map was constructed using the mTORC1/DEPTOR monomer by superimposing two copies of the monomer density onto the dimer density refined with C1 symmetry in Relion 3D. A composite map with the two mTORC1 monomer maps was calculated using EMDA (*Warshamanage and Murshudov, 2021*). The symmetrized dimer model was adjusted manually and refined with REFMAC5 and ISOLDE to resolve a few minor clashes at the interface between the monomers. One monomer was extracted and submitted to PDB with BIOMT records to reconstruct the biological dimeric assembly. Local resolutions were estimated using ResMap (*Figure 3—figure supplement 2E,G*).

For the A1459P mutant mTORC1 in a complex with DEPTOR, the wild-type monomer mTORC1/DEPTOR was used as an initial model and placed into a monomer density in the A1459P mTORC1 cryo-EM map with C2 symmetry. The model was adjusted manually and refined using ISOLDE. The ISOLDE-refined monomer was further refined in REFMAC5 using the C2 dimer density, with strict C2 symmetry constraints and with the wild-type monomer for external restraints.

The density for the PDZ domain was clearer in the mTORC1-A1459P/DEPTOR structure than it was in the wild-type structure, probably reflecting a greater affinity of this activated construct for the DEPTOR PDZ. To obtain an unbiased verification of the placement of the PDZ domain we masked out all density due to mTORC1, leaving only density in the region between the FAT domain and the N-heat, and this density was about the size that we would expect for the PDZ domain. We used Chimera to do a global search of this density to find the optimal translation and orientation of the PDZ

model in the density. We placed the NMR model in 100,000 initial orientations/translations and carried out a local search for each placement. The initial placements were random orientations and random translations, with the translations restricted so that the center of the PDZ model would be within 10 Å of the center of the putative PDZ density. The orientation/translation that yielded the highest correlation coefficient (0.86) corresponded to the placement that we chose when manually fitting the density. Among 100,000 trials carried out in the global search, this placement was located 4873 times. It is likely that this represents the global maximum correlation placement of the NMR model in the cryo-EM density, since the next most common placement had a lower correlation (0.82) and was reached only 874 times. The lowest correlation coefficient observed in the broad search was 0.73 (found one time in 100,000 trials). The results suggest that our model of the PDZ interaction with mTORC1 is unique and optimal.

## Multibody refinement analysis of wild-type mTORC1/DEPTOR particles

Multibody refinement analysis was performed following the protocol described previously (*Nakane et al., 2018*). The wild-type mTORC1/ DEPTOR complex was split into four bodies of >100 kDa each via the Segger tool in Chimera (*Figure 6—figure supplement 1B*; *Pintilie et al., 2010*). They were arbitrarily named 'mTOR', 'RAPTOR', 'M-heat', and 'N-heat', although they do not correspond to these exact domain boundaries. Corresponding masks were created using 15 Å lowpass filtered maps and soft-edges with a width of 20 Å. All bodies were set to rotate relative to the 'mTOR' body by 10 or 15 degree, 'mTOR' was set to rotate relative to the 'M-heat', and the standard deviations on the body translations were all set to 2 to 3 pixels (2.8–3.2 Å). With the relion_flex_analyse program two separate STAR files with 10,558 particles and 13,975 particles were created, for which the amplitude along the third eigenvector is less than −7, or greater than 7, respectively (*Figure 6—figure supplement 1B*). Separate refinements of these subsets yielded maps with overall resolution estimates of 8.3 and 7.6 Å, respectively.

## NMR sample preparation

For NMR experiments, isotopically labeled DEPTOR PDZ and linker-PDZ were expressed in *E. coli* C41(DE3)-RIPL cells with 100 µg/mL ampicillin. After overnight growth in 50 mL 2xTY media, 20 mL of the starter culture was spun down at 3000 g and washed with water. The cell pellet was resuspended in 1 L M9 minimal media (6 g/l $Na_2HPO_4$, 3 g/l $KH_2PO_4$, 0.5 g/l NaCl) with 1.7 g/L yeast nitrogen base without amino acids and ammonium sulfate (YNB - Sigma Y1251). 1 g/L $^{15}NH_4Cl$ and 4 g/L $^{13}C$-glucose were supplemented for $^{15}N$ and $^{13}C$ labeling, respectively. Cells were grown at 220 rpm in a shaking incubator and expression were induced with 1 mM IPTG at $OD_{600}$ = 0.8 at 23°C for 16 hr.

To produce perdeuterated protein, cells were adapted for growth in deuterated M9 minimal media with 1.7 g/L YNB on agar plates containing 10%, 44% and 78% $D_2O$, before switching to 100% perdeuterated media in a 50 mL starter culture. The starter culture was then innoculated into 1 L M9 minimal media prepared in 100% $D_2O$, supplemented with 1.7 g/L YNB, 1 g/L $^{14}NH_4Cl$ and 4 g/L $^2H^{13}C$-glucose. Cells were grown at 220 rpm in a shaking incubator and expression were induced with 1 mM IPTG at $OD_{600}$ = 0.8 at 25°C for 18 hr.

The isolated mTOR FRB domain (residues 2015–2114) was purified from a 6 L culture of C41(DE3) RIPL cells transformed with plasmid pOPL107, grown to $OD_{600}$ = 0.8 and induced with 0.3 mM IPTG at 16°C for 18 hr. Cells were lysed by sonication in a GST-A buffer (50 mM Tris pH 8, 100 mM NaCl, 1 mM TCEP) supplemented with 0.25 mg/mL lysozyme and 2 µL/100 mL of Universal nuclease. Following ultracentrifugation at 35 k rpm in Ti45 rotor for 35 min, the supernatant was purified by affinity chromatography on the Glutathione-Sepharose 4B beads (GE Healthcare, GE17-0756-05) equilibrated in GST-A buffer. The GST-FRB fusion was eluted with 20 mM glutathione in the same buffer. The eluate was diluted with two volumes of dilution buffer (50 mM Tris pH 8, 1 mM TCEP), loaded on a 5 mL HiTrapQ column and eluted with a 0–1M NaCl gradient. The fractions containing GST-FRB were concentrated in a 30K Ultra-15 concentrator and further purified by gel filtration on a Superdex 75 16/60 column equilibrated in 50 mM HEPES pH 8, 100 mM NaCl, 1 mM TCEP.

The DEPTOR linker (residues 228–323) was purified from a 6 L culture of C41(DE3)RIPL cells transformed with plasmid pOPL159, grown to $OD_{600}$ = 0.8 and induced with 0.3 mM IPTG at 16°C for 18 hr. The $His_6$-tagged protein was purified by affinity chromatography on Ni-NTA agarose beads

(Qiagen 30230) and cleaved on beads with TEV protease o/n at 4°C. The cleaved protein was diluted with 1 vol of 50 mM Tris pH8, 1 mM TCEP (to achieve ~ 50 mM NaCl) and loaded on a 5 mL HiTrapQ column equilibrated in 50 mM HEPES pH8, 25 mM NaCl, 1 mM TCEP. The flow-through fraction containing DEPTOR linker was concentrated in a 3K Amicon Ultra-15 concentrator and further purified by gel filtration on Superdex 75 16/60 column equilibrated in 50 mM HEPES pH 8, 100 mM NaCl, 1 mM TCEP.

## DEPTOR PDZ homology model generation by NMR

All NMR data sets were collected at 278K using Bruker Avance II+ 700 MHz or 600MHz Avance III spectrometers with TCI triple resonance cryoprobes unless otherwise stated. All samples were prepared with 5% $D_2O$ as a lock solvent, at pH 8 with 50 mM HEPES and 200 mM NaCl.

$^1$H-$^{15}$N BEST-TROSY (band selective excitation short transients-transverse relaxation optimized spectroscopy) were collected for all samples using an optimized pulse sequence (*Favier and Brutscher, 2011*). The assignment of backbone $H_N$, N and Cα,Cβ resonances of the 190 μM $^{13}$C/$^{15}$N DEPTOR PDZ (residues 324–409) sample was completed using the following 3D datasets acquired as pairs to provide own and preceding carbon connectivities. In most cases the amide proton and nitrogen dimension were taken from the $^1$H-$^{15}$N BEST-TROSY experiment: HNCO, HN(CA)CO, CBCA(CO)NH and HNCACB, which were used as experimental pairs with 1024, 64 and 110 complex points in the proton nitrogen and carbon dimensions, respectively. Partial Hα, Hβ side-chain chemical shift assignments were obtained from an HBHA(CO)NH spectrum collected with 1024, 64, and 110 complex points in the proton, nitrogen and the second proton dimensions, respectively. Assignment of the carbon side-chain resonances was completed with HC(C)H- and (H)CCH-TOCSY experiments (also collected with 1024, 64, and 110 points). These assignments enabled the analysis of a limited set of through space connectivities from $^{15}$N and $^{13}$C edited NOESY experiments both acquired with a mixing time of 120 ms and collected with 2048, 60/80, and 160 points in the proton, nitrogen/carbon, and second proton dimensions, respectively. All data were processed using Topspin 3.1 (Bruker, RRID:SCR_014227) and analyzed using NMRFAM-Sparky (*Lee et al., 2015*, RRID: SCR_014228). The backbone assignment was aided using MARS (*Jung and Zweckstetter, 2004*). The assigned backbone chemical shifts were used to guide calculation of a structural model of the PDZ domain using POMONA/CS-RosettaCM (*Shen and Bax, 2015*). Including a limited number of long-distance NOE restraints further refined the model; these restraints were curated so that they were from amino acids three or more residues apart as previously described (*Raman et al., 2010*).

## NMR dynamics characterization and binding experiments

The dynamic properties of the DEPTOR PDZ protein were investigated using standard Bruker $^{15}$N $T_1$, $T_2$ and $^{15}$N{$^1$H}NOE [heteronuclearNOE] experiments. $T_1$ relaxation times were calculated using delays of 10, 20, 40, 80, 160, 320, 640, 1280, and 2000 ms and $T_2$ relaxation times with delays of 16.9, 33.8, 67.6, 101.4, 135.2, 169.0, 202.8, and 253.5 ms, collected with 3 s relaxation delays. Peak intensities and curve fitting were calculated using Sparky. The heteronuclearNOE experiment was collected as a pseudo 3D spectrum, using a 120° proton pulse train with a 5 ms delay for a total of 5 s, with interleaved on and off resonance saturation. The hetNOE values were calculated from peak intensities according to the equation $I_{on}/I_{off}$.

To observe the interaction of DEPTOR PDZ with mTORC1, 32 μM of $^2$H,$^{13}$C,$^{15}$N DEPTOR PDZ was added to 3.2 μM of the mTORC1 complex, with the excess PDZ shifting the equilibrium towards a higher percentage of bound state. Here, binding is observed as a residual effect in the unbound pool of PDZ and is manifested as line broadening in the $^1$H-$^{15}$N HSQC experiment when compared to free DEPTOR PDZ only. The $^{15}$N-$^1$H HSQC experiment was used here instead of the $^1$H-$^{15}$N BEST-TROSY to avoid potential solvent exchange bias. Peaks heights were normalized to the signal of the C-terminal residue before the ratio calculated. Peaks that had reduced relative intensity define the interaction surface for the PDZ domain.

Structural differences between the DEPTOR PDZ domain and the DEPTOR linker-PDZ construct were first identified by chemical shift perturbations in $^1$H-$^{15}$N BEST-TROSY experiments. In the absence of a complete linker-PDZ assignment each signal of the linker-PDZ spectrum from a 114 μM sample was compared with that of the assigned PDZ domain collected under the same conditions, giving a weighted chemical shift perturbation calculated by *Amin et al., 2013*; *Rowe et al., 2009*:

$$\Delta\delta\text{total} = \left(\delta H^2 + \left(\frac{\delta N}{5}\right)^2\right)^{0.5}$$

with the smallest perturbation reported as a minimal chemical shift perturbation map (*Muskett et al., 1998*; *Williamson et al., 1997*).

Both the mTOR-FRB domain and the DEPTOR linker construct were assigned in order to identify residues involved in the binding interaction. FRB assignments were obtained at 293K and transferred to the 278K spectra using a temperature titration. The backbone assignment of the 70 µM FRB sample (residues 2015–2114) was completed using 3D HNCO, HN(CA)CO and HNCACB, CBCA(CO)NH experimental pairs - all collected with 1024, 64, and 96 points in the proton, nitrogen and carbon dimensions respectively and with 20–40% non-uniform sampling (NUS). Data were processed using NMRPipe (*Delaglio et al., 1995*) including compressed sensing for data reconstruction (*Kazimierczuk and Orekhov, 2011*) and analyzed as above for the PDZ data. A 140 µM $^{15}$N/$^{13}$C DEPTOR linker (228-323) construct was assigned using the same experiment pairs as above, supplemented with HN(COCA)NNH and HN(CA)NNH experiments (2048, 64, and 80 points in the proton, nitrogen, and carbon dimensions, and 25% NUS) to provide sequential N,N connectivities. Carbon-detect 3D experiments, in recent years established for the assignment of disordered proteins due to their superior resolution, allowed the completion of the sequential assignment including proline connectivities. Carbon-detect 3D experiment pairs (HCA)CON and (HCA)NCO were collected with up to 1024, 128, and 80 complex points in the carbonyl-carbon, nitrogen, and indirect aliphatic-carbon dimensions respectively, whereas the CBCACON and CBCANCO pair was acquired with 1024, 72, and 80 points in carbonyl-carbon, nitrogen, and indirect aliphatic-carbon dimensions.

Secondary chemical shift analysis to describe conformational preferences for the DEPTOR linker was based on a comparison of assigned backbone carbon α and β shifts with chemical shift values expected for a random coil protein with the same sequence and under the same experimental conditions (pH and temperature), calculated using methods described previously (*Kjaergaard et al., 2011*; *Kjaergaard and Poulsen, 2011*; *Schwarzinger et al., 2001*). Random coil values were subtracted from the experimental data to give ΔσCα and ΔσCβ, with ΔσCα-ΔσCβ plotted against residue number.

Binding of the DEPTOR linker to mTORC1 FRB was observed by $^1$H-$^{15}$N BEST-TROSY NMR. FRB residues involved in the binding were identified by the addition of up to 220 µM of unlabeled linker-PDZ to 40 µM of $^{15}$N-labeled FRB. Similarly, DEPTOR linker residues involved in binding were identified by the addition of up to 320 µM of unlabeled FRB to a 40 µM $^{15}$N-labeled DEPTOR linker sample. Data from both titrations were analyzed using the above equation.

## HDX experiments

Experiments followed suggested standards by the HDX-MS community (*Masson et al., 2019*). For global exchange, a 5 µL solution of 5 µM DEPTOR (50 mM HEPES pH 8.0, 100 mM NaCl, 1 mM TCEP) was incubated for 3 s with 40 µL of ice-cold D$_2$O buffer of identical composition (92.8% D$_2$O), for a final concentration of 74.24% D$_2$O. The reaction was quenched using 20 µL of 2 M guanidinium chloride and 2.4% v/v formic acid, shock-frozen in liquid nitrogen and stored at −80°C. For measurement of the incorporation of deuterium, samples were thawed and injected onto an M-Class Acquity UPLC with HDX Manager technology (Waters) kept at 0.1°C. Samples were digested on-line using an Enzymate BEH immobilized Pepsin Column (Waters, 186007233) at 15°C for two min, with peptides being eluted onto an Acquity UPLC BEH C18 column (Waters, 186002346), equilibrated in Buffer Pepsin-A (0.1% v/v formic acid), using a 3–43% gradient of Pepsin-B buffer (0.1% v/v formic acid, 99.9% acetonitrile) over 12 min. Data were collected on a Waters Synapt G2 Si using MS$^e$ mode (*Silva et al., 2005*), using an electrospray source (set at 3 kV), from 50 to 1800 m/z. Peptides were identified from non-deuterated samples of DEPTOR in the independent replicates using ProteinLynx Global Server (Waters, RRID:SCR_016664) against a library of DEPTOR and porcine pepsin, and then imported into DynamX (Waters) software using the following criteria for automated selection: Minimum Intensity 7500, minimum sequence length 6, maximum sequence length 30, minimum products 1, minimum products per amino acid 0.11, minimum consecutive products 1, minimum sum Intensity for products 500, maximum MH+ error five ppm, and identification in all three non-deuterated files.

205 peptides were initially identified, reduced to 179 after a manual quality control process (presence of overlapping peptides *etc.*) was conducted.

The HDX binding study of FRB and DEPTOR was performed by preincubating 100 µM FRB with 100 µM DEPTOR in buffer (50 mM HEPES pH 8.0, 100 mM NaCl, 1 mM TCEP) at room temperature for 1 hr. An aliquot of 5 µL was incubated with 45 µL of $D_2O$ buffer at room temperature for 3, 30, 300, and 3000 s, the reaction was quenched and treated as described, with the exception of using a 5–36% gradient of acetonitrile in 0.1% v/v formic acid for elution from Acquity UPLC BEH C18 column. Data were collected from 300 to 2000 m/z, and mass analysis was performed as described above. Deuterium incorporation was not corrected for back-exchange and represents relative, rather than absolute changes in deuterium levels. Changes in H/D amide exchange in any peptide may be due to a single amide or a number of amides within that peptide. All time points in this study were prepared at the same time and individual time points were acquired on the mass spectrometer on the same day.

## Surface plasmon resonance

Twin-Strep-tagged wild-type and activated mTORC1 mutant A1459P were purified as described above with the exception that no TEV-protease was added, and the protein was eluted from the Strep-Trap HP columns using 10 mM desthiobiotin in 40 mL elution buffer prior loading onto a 5 mL HiTrap Q column.

SPR was performed using a Biacore T200 using CM5-sensor chips (Cytiva). Both reference control and analyte channels were equilibrated 50 mM HEPES pH 7.5, 100 mM NaCl, 1 mM TCEP. Twin-Strep-tagged mTOR was captured onto a Strep-Tactin XT (IBA Lifesciences) coated surface prepared according to the supplied instructions. SPR runs were performed with analytes injected for 120 s followed by a 300 s dissociation in a 1:2 dilution series with initial concentrations of 20 µM for DEPTOR PDZ (residues 305–409). After reference and buffer signal correction, sensogram data were fitted using GraphPad Prism (RRID:SCR_002798). The equilibrium response ($R_{eq}$) data were fitted to: a single site interaction model to determine $K_d$:

$$R_{eq} = \left( \frac{CR_{max}}{C + K_d} \right) + N_s C + B$$

where C is the analyte concentration and $R_{max}$ is the maximum response at saturation, $N_s$ is a linear non-specific binding term and B is the background resonance; or a two-site model:

$$R_{eq} = \left( \frac{CR_{max1}}{C + K_{d1}} \right) + \left( \frac{CR_{max2}}{C + K_{d2}} \right) + B$$

where $R_{max1}$ and $K_{d1}$, and $R_{max2}$ and $K_{d2}$ are the maximum response and dissociation constants for site 1 and 2 respectively.

# Acknowledgements

We acknowledge the MRC - Laboratory of Molecular Biology Electron Microscopy Facility for access and support of electron microscopy sample preparation and data collection. We thank Christos Savva, Rangana Warshamanage, Domagoj Baretić, Xiao-chen Bai, Rafael Fernández-Leiro and Sjors Scheres for expert assistance with cryo-EM data collection and processing. We thank Jake Grimmet and Toby Darling for implementing and maintaining the scientific computing infrastructure at the MRC LMB. We thank Christopher Johnson for training for the differential scanning fluorimetry and maintaining the Biophysics facility at the MRC LMB. We thank Yohei Ohashi for help with the initial mTORC1 purification. We thank Jaslyn Wong for providing purified mTORC1 A1459P and RHEB. MA was supported by FEBS fellowship and EMBO Advanced Fellowship (EMBO ALTF 603–2019). The work was supported by the Medical Research Council (MC_U105184308 to RLW) and Cancer Research UK (grant C14801/A21211 to RLW).

## Additional information

### Funding

| Funder | Grant reference number | Author |
|---|---|---|
| Medical Research Council | MC_U105184308 | Roger L Williams |
| Cancer Research UK | C14801/A21211 | Roger L Williams |
| EMBO | EMBO ALTF 603–2019 | Madhanagopal Anandapadamanaban |
| FEBS | Fellowship | Madhanagopal Anandapadamanaban |

The funders had no role in study design, data collection and interpretation, or the decision to submit the work for publication.

### Author contributions

Maren Heimhalt, Alex Berndt, Conceptualization, Formal analysis, Validation, Investigation, Visualization, Methodology, Writing - original draft, Writing - review and editing; Jane Wagstaff, Madhanagopal Anandapadamanaban, Formal analysis, Validation, Investigation, Visualization, Methodology, Writing - review and editing; Olga Perisic, Resources, Investigation, Methodology, Writing - review and editing; Sarah Maslen, Stephen McLaughlin, Glenn R Masson, Andreas Boland, Xiaodan Ni, Formal analysis, Investigation, Methodology, Writing - review and editing; Conny Wing-Heng Yu, Investigation, Methodology, Writing - review and editing; Keitaro Yamashita, Software, Formal analysis, Investigation, Methodology, Writing - review and editing; Garib N Murshudov, Software, Formal analysis, Supervision, Methodology, Writing - review and editing; Mark Skehel, Formal analysis, Supervision, Investigation, Methodology, Writing - review and editing; Stefan M Freund, Supervision, Investigation, Methodology, Writing - review and editing; Roger L Williams, Conceptualization, Resources, Formal analysis, Supervision, Funding acquisition, Visualization, Project administration, Writing - review and editing

### Author ORCIDs

Maren Heimhalt (ID) https://orcid.org/0000-0001-9993-2884
Madhanagopal Anandapadamanaban (ID) https://orcid.org/0000-0002-4237-0048
Stephen McLaughlin (ID) https://orcid.org/0000-0001-9135-6253
Roger L Williams (ID) https://orcid.org/0000-0001-7754-4207

### Decision letter and Author response

Decision letter https://doi.org/10.7554/eLife.68799.sa1
Author response https://doi.org/10.7554/eLife.68799.sa2

## Additional files

### Supplementary files

• Supplementary file 1. Global HDX-MS measurements for the full-length, human DEPTOR.

• Supplementary file 2. Summary of HDX-MS results for DEPTOR in the presence and absence of the mTOR-FRB domain, related to *Figure 5C*.

• Transparent reporting form

### Data availability

The cryo-EM map and the model are deposited with the EMDB and PDB, respectively. Backbone assignments of DEPTOR PDZ, mTOR-FRB and DEPTOR linker have been submitted to the BMRB, Biological Magnetic Resonance Bank with the accession numbers 50324, 50325 and 50326, respectively.

The following datasets were generated:

| Author(s) | Year | Dataset title | Dataset URL | Database and Identifier |
|---|---|---|---|---|
| Heimhalt M, Berndt A, Wagstaff J, Anandapadamanaban M, Perisic O, Maslen S, McLaughlin S, Yu CW-H, Masson GR, Boland A, Ni X, Yamashita K, Murshudov GN | 2021 | human DEPTOR in a complex with mutant human mTORC1 A1459P | https://www.ebi.ac.uk/pdbe/entry/pdb/7OWG | RCSB Protein Data Bank, 7OWG |
| Heimhalt M, Berndt A, Wagstaff J, Perisic O, Maslen S, Yu CWH, Anandapadamanaban M, Masson GR, Boland A, Johnson C, McLaughlin S, Skehel M, Freund SM, Williams RL | 2021 | Assigned backbone chemical shifts of DEPTOR-PDZ | https://bmrb.io/data_library/summary/index.php?bmrbId=50324 | Biological Magnetic Resonance Data Bank, 50324 |
| Heimhalt M, Berndt A, Wagstaff J, Perisic O, Maslen S, Yu CWH, Anandapadamanaban M, Masson GR, Boland A, Johnson C, McLaughlin S, Skehel M, Freund SM, Williams RL | 2021 | Backbone chemical shift assignment of the FRB domain of mTOR | https://bmrb.io/data_library/summary/index.php?bmrbId=50325 | Biological Magnetic Resonance Data Bank, 50325 |
| Heimhalt M, Berndt A, Wagstaff J, Perisic O, Maslen S, Yu CWH, Anandapadamanaban M, Masson GR, Boland A, Johnson C, McLaughlin S, Skehel M, Freund SM, Williams RL | 2021 | Backbone chemical shift assignment of the linker region of DEPTOR | https://bmrb.io/data_library/summary/index.php?bmrbId=50326 | Biological Magnetic Resonance Data Bank, 50326 |
| Heimhalt M, Berndt A, Anandapadamanaban M, Perisic O, Maslen S, McLaughlin S, Yu CWH, Masson GR, Boland A, Yamashita K, Murshudov GN, Skehel M, Freund SM, Williams RL | 2021 | 7OX0 | https://www.ebi.ac.uk/pdbe/entry/pdb/7OX0 | Protein Data Bank, 7OX0 |

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
