## [Decision Letter]

**Acceptance summary:**

DEPTOR is a regulator of the central mTORC1 and mTORC2 kinase complexes that is both of general interest to biologists and has remained poorly understood despite many years of investigation. Two *eLife* manuscripts report new structural insights into DEPTOR's mechanism of action that will be of broad impact for cell biologists, kinase enzymologists and mTORC1/mTORC2 specialists.

**Decision letter after peer review:**

Thank you for submitting your article "Bipartite binding and partial inhibition links DEPTOR and mTOR in a mutually antagonistic embrace" for consideration by *eLife*. Your article has been reviewed by 3 peer reviewers, including Adam Frost as the Reviewing Editor and Reviewer #1, and the evaluation has been overseen by Volker Dötsch as the Senior Editor. The following individual involved in the review of your submission has agreed to reveal their identity: Roberto Zoncu (Reviewer #2).

Essential revisions:

1. mTORC1 phosphorylates DEPTOR in the linker region, and the authors show that DEPTOR pre-phosphorylated by activated mTORC1 (phospho-DEPTOR) does not inhibit mTORC1. Hence, the summary model shown in Figure 7B proposes "mTORC1 activity alone is sufficient to reduce DEPTOR inhibition" through direct phosphorylation of DEPTOR's linker. Does this model predict that the 13A mutant constructs of DEPTOR that are not phosphorylated should be more potent inhibitors of mTORC1 – because these constructs will not be subject to phosphorylation-dependent negative feedback? The authors show that the 13A constructs are inhibitors with similar properties to the wild-type sequence. Please comment.

2. Understanding the structural basis of DEPTOR linker binding to the FRB domain of mTORC1 is critical to explaining why the longer linker is necessary for DEPTOR's partial inhibition. The NMR experiments to corroborate the cryoEM density interpretation, however, raise questions. First, regarding the chemical shift potentials (CSPs), please cite the basis for the weighting term. It appears the authors did not divide by two before taking the square root. Does omitting this step increase the apparent CSPs (which are already small)? Related, is there a precedent for using "minimal" CSP maps? If so, please include citations. I am not an expert but it seems dubious to calculate CSPs compared to unassigned spectra. Also, please annotate the CSP figure legends to indicate the "Max" std value for the scale since the CSPs for the PDZ domain appear to be more robust than for the linker.

3. In Figure 5—figure supplement 1, please overlay the red on top of the blue and zoom in on a few data callouts in detail, showing fewer contour levels. As currently depicted, it is challenging to appreciate peak shifts. Also, DEPTOR linker residue 262 is highlighted in the Figure 5B comparison – but is it aliased with F261, F290, and V309? I can't tell from the figure if the peak centers are distinct enough, and I believe aliased peaks should be excluded from the CSP analysis.

4. Figure 5C is confusing. The FRB domain of mTORC1 appears to protect surfaces from solvent exchange throughout full-length DEPTOR (and some regions more strongly than the putative linker binding region).

5. It is concerning that the cryo-EM reconstruction of the A1459P mTORC1 mutant bound to DEPTOR failed to recover any putative linker density bound to the FRB helices (Figure 6C). Integrating all of the above, and in the absence of structure-based point mutants that impair linker binding to the FRB and partial kinase inhibition, the mechanism by which the linker tunes mTORC1 activity remains unclear.

6. In several panels, the phosphor signal is too saturated to appreciate the extent of DEPTOR-dependent inhibition (e.g. 1A, 2B). Shorter exposures should be shown, instead of or along with the current blots.

7. In the section titled "Cryo-EM structure of mTORC1/DEPTOR reveals a bipartite binding mode of DEPTOR to mTOR," the authors indicate specific residues on the PDZ domain of DEPTOR that bind to mTORC1's FAT domain. The placement of this statement suggests that this information is garnered from docking the PDZ NMR structure into the cryoEM map. From the data presented, however, docking the PDZ domain unambiguously into the density of the cryoEM map seems challenging due to the poor resolution of the peripheral densities. Please comment.

8. In figure 2C it would be useful to see the WT DEPTOR plot. The western blot appears to show higher inhibition is achieved with the Linker+PDZ 13A domain as compared to WT DEPTOR or DEPTOR 13A. Do the authors have any models of what would give rise to this heightened effect with the minimal domain?

---

## [Author Response]

Essential revisions:1. mTORC1 phosphorylates DEPTOR in the linker region, and the authors show that DEPTOR pre-phosphorylated by activated mTORC1 (phospho-DEPTOR) does not inhibit mTORC1. Hence, the summary model shown in Figure 7B proposes "mTORC1 activity alone is sufficient to reduce DEPTOR inhibition" through direct phosphorylation of DEPTOR's linker. Does this model predict that the 13A mutant constructs of DEPTOR that are not phosphorylated should be more potent inhibitors of mTORC1 – because these constructs will not be subject to phosphorylation-dependent negative feedback? The authors show that the 13A constructs are inhibitors with similar properties to the wild-type sequence. Please comment.

The reviewer has raised a point that could cause some confusion and we have tried to clarify it. Phosphorylated DEPTOR is not an inhibitor of mTORC1. However, our assays were carried out in steady state conditions. This means that we are measuring initial rates during which there is no appreciable depletion of substrate and the appearance of product is linear with time. Under these conditions, there will be very little appearance of phosphorylated DEPTOR relative to unphosphorylated DEPTOR. It is for this reason that WT and the 13 S/T-A mutant of DEPTOR have similar steady-state properties. Under these conditions, the WT DEPTOR is not significantly phosphorylated. DEPTOR’s characteristics as an mTORC1 substrate are shown in Figure 2—figure supplement 1. This shows that after 15 min only a small fraction of DEPTOR is phosphorylated. Assays of inhibition of DEPTOR constructs by mTORC1-A1459P were performed over a 4 min time interval. In order to observe the loss of inhibition due to phosphorylation of DEPTOR, we had to pre-phosphorylate DEPTOR by incubating it with mTORC1 for a longer time. In order to circumvent any potential confusion, we have expanded our explanation regarding this point in the revised manuscript on page 17.

2. Understanding the structural basis of DEPTOR linker binding to the FRB domain of mTORC1 is critical to explaining why the longer linker is necessary for DEPTOR's partial inhibition. The NMR experiments to corroborate the cryoEM density interpretation, however, raise questions. First, regarding the chemical shift potentials (CSPs), please cite the basis for the weighting term. It appears the authors did not divide by two before taking the square root. Does omitting this step increase the apparent CSPs (which are already small)? Related, is there a precedent for using "minimal" CSP maps? If so, please include citations. I am not an expert but it seems dubious to calculate CSPs compared to unassigned spectra. Also, please annotate the CSP figure legends to indicate the "Max" std value for the scale since the CSPs for the PDZ domain appear to be more robust than for the linker.

The Euclidian weighing term used in the NMR analysis is from the following references that we have added to the manuscript (Amin et al., 2013; Rowe et al., 2009). There are a number of different ways to report weighted chemical shifts in the literature (Shuker et al., 1996), and here we report the Euclidian distances as opposed to the average as suggested by Williamson (Williamson, 2013). There is a precedent in the literature for the minimal shift maps, and we have added these references to the manuscript (Muskett et al., 1998; Williamson et al., 1997). We feel that the minimal shift map is an important tool for describing differences between states where one state is not assignable. The complex spectra of the DEPTOR long linker-PDZ could not be assigned due to the different dynamic qualities of the folded and disordered parts of the construct. As the assignment of the PDZ alone is known, it is reasonable to assume that peaks shifted in relation to the long-linker PDZ protein are shifted as a result of a change in the local environment as a consequence of the presence of the long linker. The minimal map identifies the likely peak locations in the unassigned spectrum in a conservative approach that can only underestimate the maximal chemical shift perturbation. It is true that line broadened peaks are not identified in the minimal map, however, this analysis will highlight line-broadened peaks and as a result residues that are altered in the new sample conditions.

The analysis of CSP values in absolute terms in not straightforward due to the nature of the residues involved, and as a result, comparing the extent of CSP between systems is highly problematic. With this is mind we have not added the “Max” Std values to the CSP legends.

3. In Figure 5—figure supplement 1, please overlay the red on top of the blue and zoom in on a few data callouts in detail, showing fewer contour levels. As currently depicted, it is challenging to appreciate peak shifts. Also, DEPTOR linker residue 262 is highlighted in the Figure 5B comparison – but is it aliased with F261, F290, and V309? I can't tell from the figure if the peak centers are distinct enough, and I believe aliased peaks should be excluded from the CSP analysis.

We have altered the figure as suggested, and included a couple of zoomed in peaks in the figure. The zoom that shows residue M262 highlights that the peak position is unambiguous and so can be included in the analysis.

4. Figure 5C is confusing. The FRB domain of mTORC1 appears to protect surfaces from solvent exchange throughout full-length DEPTOR (and some regions more strongly than the putative linker binding region).

Our HDX-MS results show that in addition to the long-linker, the N-terminal region of DEPTOR (peptide 2-32) shows substantial protection from solvent exchange in the presence of the FRB domain of mTORC1. We did not comment on the changes in this region, since we know that they are not essential for either the DEPTOR-mediated inhibition of mTORC1 or the partial inhibition unique to DEPTOR. However, we can now provide a structural hypothesis for the origin of these changes in protection in the N-terminus of DEPTOR, since while our manuscript was in review, another manuscript describing the structure of the tandem DEP domains at the N-terminus of DEPTOR was published (Weng et al., JMB; 2021, PMID 33865870). In this structure, a short segment (228-235) from the beginning of what we refer to as the DEPTOR long linker (228-323) forms a short helix that interacts with residues in a segment 20-32 at the N-terminus of the first DEP domain. This N-terminal region becomes more protected upon interaction with the mTOR FRB. It is plausible that the FRB interacting with the long linker promotes ordering of the linker and thereby protection of the N-terminal 2-32 peptide, which contacts the long linker. This also might account for the lower IC50 of the full-length DEPTOR compared with the construct lacking the tandem DEP domains (Figure 2C). We have now briefly commented on this in the text on page 14.

5. It is concerning that the cryo-EM reconstruction of the A1459P mTORC1 mutant bound to DEPTOR failed to recover any putative linker density bound to the FRB helices (Figure 6C). Integrating all of the above, and in the absence of structure-based point mutants that impair linker binding to the FRB and partial kinase inhibition, the mechanism by which the linker tunes mTORC1 activity remains unclear.

We have a reasonable explanation for the absence of the DEPTOR linker density at the FRB for the mTORC1-A1459P construct bound to DEPTOR. The mTORC1-A1459P/DEPTOR sample was not cross-linked. When we do not cross-link, the apparent helical density bound to the FRB is not evident. This is not especially surprising given the weak affinity of this FRB site for DEPTOR and other substrates. The original cryo-EM study of the mTORC1 complex by Pavletich and his colleagues (Yang et al., Nature 2017) also showed no density on the FRB in the presence of the substrate 4EBP1, although the kinetic results in the same study show the importance of 4EBP1 interaction with the FRB for maximal rate. The structural demonstration of this interaction with the FRB was shown by fusing the substrate to the FRB and crystallising the fusion FRB construct (Yang et al., Nature 2017). The fusion construct was one way to capture experimentally the interaction. In our work with wild-type DEPTOR, we have captured this interaction by crosslinking. Although this is a weak interaction, it is a classic observation that weak interactions can have profound influences on the overall affinity of protein/protein interactions. This is clear from the thermodynamics of independent interactions, and there is strong evidence that multiple weak interactions may be preferred in biological systems to impart greater specificity. This has been nicely demonstrated by the recent work of Scheepers et al. (PNAS 117, 22690–22697; 2020, PMID 32859760).

It is not clear what the reviewer means by the statement “Integrating all of the above…” Our NMR study characterises the interaction of the DEPTOR linker with the FRB. The HDX-MS is also consistent with this interaction. We did not make point mutations in the DEPTOR linker, because it is clear from the NMR study that there are multiple regions in the linker that can interact with the FRB. We did, however, delete the linker and show that we lose all inhibition. We did not make mutations in the FRB, because it has been shown previously that this site is important for substrate interaction. Mutagenesis would have been far less informative than the extensive work we did to characterise the interactions between DEPTOR and the FRB.

6. In several panels, the phosphor signal is too saturated to appreciate the extent of DEPTOR-dependent inhibition (e.g. 1A, 2B). Shorter exposures should be shown, instead of or along with the current blots.

This was a good suggestion and the figures 1 and 2 have been revised. Both the long and short exposures are now presented, and this makes our conclusions more visually obvious.

7. In the section titled "Cryo-EM structure of mTORC1/DEPTOR reveals a bipartite binding mode of DEPTOR to mTOR," the authors indicate specific residues on the PDZ domain of DEPTOR that bind to mTORC1's FAT domain. The placement of this statement suggests that this information is garnered from docking the PDZ NMR structure into the cryoEM map. From the data presented, however, docking the PDZ domain unambiguously into the density of the cryoEM map seems challenging due to the poor resolution of the peripheral densities. Please comment.

The reviewer is correct. We garnered our initial assessment of residues interacting with the PDZ domain by examining the residues in the model that were closest to the mTOR after fitting the NMR-derived model of the PDZ domain into the cryo-EM density. Subsequent examination of CSP and line-broadening were consistent with this initial estimation. Although the resolution of the cryo-EM density is limited, the fit of the PDZ domain to the density is unique. To show this, we created a new map for the A1459P mutant in which we masked out all density due to mTORC1. This left density only in the region between the FAT domain and the N-heat, and this density was about the size that we would expect for the PDZ domain. We used Chimera to do a global search of this density to find the optimal translation and orientation of the PDZ model in the density. We placed the NMR model in 100000 initial orientations/translations and carried out a local search for each placement. The initial placements were random orientations and random translations, with the translations restricted so that the center of the PDZ model would be within 10 Å of the center of the putative PDZ density. The orientation/translation that yielded the highest correlation coefficient (0.86) corresponded to the placement that we chose when manually fitting the density. Among 100000 trials carried out in the global search, this placement was located 4873 times. The next most common placement had a lower correlation (0.82) and was reached only 874 times. Any random placement with the PDZ domain volume occupying the volume of the density will give a correlation coefficient that is statistically significant (different than zero), but they all have correlation less than the correct solution that is consistent with the NMR measurements. The lowest correlation coefficient observed in the broad search was 0.73 (found one time in 100000 trials). From this unbiased analysis, we believe that our model for the PDZ interaction with mTORC1 is unique and correct. We have briefly described this unbiased analysis in the methods on page 48.

8. In figure 2C it would be useful to see the WT DEPTOR plot. The western blot appears to show higher inhibition is achieved with the Linker+PDZ 13A domain as compared to WT DEPTOR or DEPTOR 13A. Do the authors have any models of what would give rise to this heightened effect with the minimal domain?

The IC50 is greater for the linker-PDZ 13A compared with full-length DEPTOR, suggesting that the truncated construct is a poorer inhibitor than the full-length construct. It is probably a poorer inhibitor because it binds more weakly to mTORC1 (perhaps due to a change in the order of the linker due to loss of its interaction with the tandem DEP domains as referred to in point 4 above). When we include the WT DEPTOR plot in Figure 2C as suggested by the reviewer (see Author response image 1), it is not clear that the maximal inhibition of the linker-PDZ construct is significantly greater than the maximal inhibition of the WT DEPTOR. Only at the maximum concentration that we could achieve in the assay is the inhibition due to the linker-PDZ greater than the WT DEPTOR, and the error bars for the two plots overlap for this point. If we show all the data from 2C and 2D in a single panel, it would more cluttered and difficult to interpret, so we believe that presenting them side-by-side is clearer.

We do not have a complete structural understanding of how partial inhibition arises. We know only that it requires the linker and that it involves interactions with the FRB. Since the same region on the FRB is also involved in interacting with substrates and the inhibitor PRAS40, we have a rough idea of what it looks like structurally. However, because substrate turnover can still take place with DEPTOR bound (the definition of partial inhibitor), we know that it does not completely overlap with the substrate interaction site. We also know it is distinct from PRAS40, which is a complete inhibitor that prevents substrate binding. Given that we do not have a full understanding of the structural origin of DEPTOR’s partial inhibition, we kept the speculations about it to a minimum. However, in response to the reviewer’s question, we have added some speculation on page 19 in the discussion.